# Long-chain anionic surfactants enabling stable perovskite/silicon tandems with greatly suppressed stress corrosion

Xinlong Wang[1,2], Zhiqin Ying [1] ✉, Jingming Zheng[1], Xin Li[1], Zhipeng Zhang[3], Chuanxiao Xiao[1,4], Ying Chen[1], Ming Wu[1], Zhenhai Yang[1], Jingsong Sun[1], Jia-Ru Xu[5], Jiang Sheng[1], Yuheng Zeng[1], Xi Yang [1] ✉, Guichuan Xing [3] & Jichun Ye [1] ✉

Despite the remarkable rise in the efficiency of perovskite-based solar cells, the stress-induced intrinsic instability of perovskite active layers is widely identified as a critical hurdle for upcoming commercialization. Herein, a long-alkyl-chain anionic surfactant additive is introduced to chemically ameliorate the perovskite crystallization kinetics via surface segregation and micellization, and physically construct a glue-like scaffold to eliminate the residual stresses. As a result, benefiting from the reduced defects, suppressed ion migration and improved energy level alignment, the corresponding unencapsulated perovskite single-junction and perovskite/silicon tandem devices exhibit impressive operational stability with 85.7% and 93.6% of their performance after 3000 h and 450 h at maximum power point tracking under continuous light illumination, providing one of the best stabilities to date under similar test conditions, respectively.

Perovskite/silicon tandem solar cell has emerged as a promising photovoltaic technology with the power conversion efficiency (PCE) soaring from 13.7 to 31.3% in just 7 years[1]. Despite impressively high PCEs, the long-term durability of the perovskite active layer remains the major obstacle for the commercialization of perovskite/silicon tandems[2]. Past reports mainly focused on developing appropriate encapsulation[3], elaborate crystallization engineering[4] and efficient defect passivation[5] to improve both the extrinsic and intrinsic stabilities of perovskites. However, akin to the "stress corrosion" in many metal[6], glass[7] and polymer[8] materials, the time dependent subcritical perovskite deterioration still occurs due to the unavoidable tensile stresses induced during the device fabrication and operation[9]. These stresses can not only microscopically weaken the lead-halide orbital coupling, which in turn alters the structure-related material properties (such as bandgap[10] and carrier dynamics[11]) and reduces the energy

barrier of phase transition, defect formation and ion migration[12,13], but also macroscopically impair the structural integrity by delamination and crack formation[14], thus accelerating the perovskite degradation and jeopardizing the device stability and reliability.

One prevalent strategy for mitigating this stress issue is employing long-chain small-molecule or polymer additives to enhance perovskite cohesive fracture energy by softening grain boundaries (GBs) and introducing film plasticity[15–17]. However, these long-chain additives can confine the perovskite grown and result in a more remarkable reduction of grain size than the short-chain counterpart[18,19]. In addition, their insulating properties always cause a penalty in the carrier extraction, and the variation in molecular weights can negatively affect the device reproducibility[20]. Moreover, most of these organic additives suffer from high volatility, temperature sensitivity and disorderliness[21], and thus can hardly retain their efficacy under high temperature and

[1]Ningbo Institute of Materials Technology and Engineering, Chinese Academy of Sciences (CAS), 315201 Ningbo, China. [2]University of Chinese Academy of Sciences, No.19(A) Yuquan Road, Shijingshan District, 100049 Beijing, China. [3]Joint Key Laboratory of the Ministry of Education, Institute of Applied Physics and Materials Engineering, University of Macau, Avenida da Universidade, Taipa 999078 Macao SAR, China. [4]Ningbo New Materials Testing and Evaluation Center CO., Ltd, Ningbo City 315201 Zhejiang Province, China. [5]Celanese (China) Holding Co., Ltd. Asia Technology and Innovation Center, 201210 Shanghai, China. ✉e-mail: yingzhiqin@nimte.ac.cn; yangx@nimte.ac.cn; jichun.ye@nimte.ac.cn

long-term light illumination[22], eventually incurring additional thermal instabilities[23]. Ionic surfactants, for their unique physicochemical properties[24], have been widely used as promising plasticizers to tune the mechanical and electrical properties of flexible and stretchable devices[25,26]. However, the effects of ionic surfactants on the basic perovskite mechanical properties have not, to the best of our knowledge, been studied, and the correlations between ionic surfactant structures and perovskite stresses are largely overlooked.

Here, we investigate the impacts of a commercially available classical long-alkyl-chain anionic surfactant (LAS), $[C_4mim]^+[C_nSO_4]^-$ ($n = 8$), on the perovskite optoelectronic and mechanical properties. During the perovskite deposition, the delicate balance of entropic and enthalpic contributions to total equilibrium free energy thermodynamically drives the LAS additives to first self-segregate as a monolayer at the solution-air interface and then spontaneously form large-sized micellar-like aggregates. The latter acts as pre-nucleation clusters to achieve rapid nucleation, while the former functions as a solvent molecular sieve to alleviate the solvent evaporation and slow down crystal growth, thus synergistically improving the perovskite crystallinity. More importantly, these LASs embrace the perovskite grains to form a glue-like scaffold that effectively eliminates stresses by reducing the Young's Modulus and thermal expansion coefficient, thus enhancing both the PCE and stability of the perovskite devices. Remarkably, the corresponding unencapsulated single-junction perovskite and dual-junction perovskite/silicon tandem devices, with the PCE of 21.6%, and 27.4%, retain 85.7% and 93.6% of their original PCE after 3000 h and 450 h (under maximum power point tracking in air), respectively, representing one of the highest lifetimes reported to date under similar conditions (Supplementary Tables 1 and 2).

## Results

### Reduction of residual stresses via LASs

We first employed a series of anionic surfactant additives (Supplementary Fig. 1a–c), featuring the 1-Butyl-3-methylimidazolium ($[C_4mim]^+$) cation and sulfate anions with different alkyl chain lengths ($[C_nSO_4]^-$, i.e., $n = 0, 1, 8$ for hydrogen, methyl and octyl sulfate, respectively), to decrease the Young's Modulus (YM) and the thermal expansion coefficient (CTE) of the perovskite films and revealed the effect of these mechanical properties on the stress of perovskite film (Supplementary Note 1). We chose the triple cation perovskite, $Cs_{0.05}(FA_{0.83}MA_{0.17})_{0.95}Pb(I_{0.82}Br_{0.18})_3$ with a bandgap of 1.63 eV, optimized for high efficiency and stability, as the active layer, and the nickel oxide ($NiO_x$) as the hole transport layer (HTL) because of its large bandgap, acceptable hole mobility, favorable energy level alignment, excellent stability, low cost, and potentially scalable deposition[27].

Figure 1a–e shows the surface morphologies and corresponding mechanical properties of the perovskite films with different additives using Peak Force quantitative nanomechanical atomic force microscopy (PFQNM-AFM) method. For the control film without additives, the modulus map shows a strong correlation to the grain structure and reveals substantial YM variations across the sample, indicating the local variations in the perovskite YMs. The spatially averaged YM has a value of 17.7 GPa, with two distinct domains represented by the different colors in YM map. The red domain with a high average YM of 26.2 GPa can be attributed to the contribution of grain intragranular (GI) regions, whereas the blue domain with a low average YM of 11.4 GPa belongs to the region near GBs (Fig. 1a), consistent with the previous report[28]. After the incorporation of $[C_4mim]^+[C_0SO_4]^-$, the perovskite film exhibits a slightly decreased grain size (Supplementary Fig. 2a, b) and a significantly increased average YM of 21.1 GPa, with high YMs of 34.5 and 14.2 GPa in GI and GB regions, respectively (Fig. 1b). These small and rigid zero-alkyl-chain additives provide a high-density of heterogeneous nucleation sites during the perovskite film formation, thus restricting the grain growth and leading to a higher perovskite macro packing density and modulus (akin to the extraordinary mechanical strength observed in nacre and fiber-reinforced polymer composites)[29]. The increased YMs can also be observed (Supplementary Fig. 3) for another short-chain $[C_4mim]^+[BF_4]^-$ additive (Supplementary Fig. 1d), which is chosen for

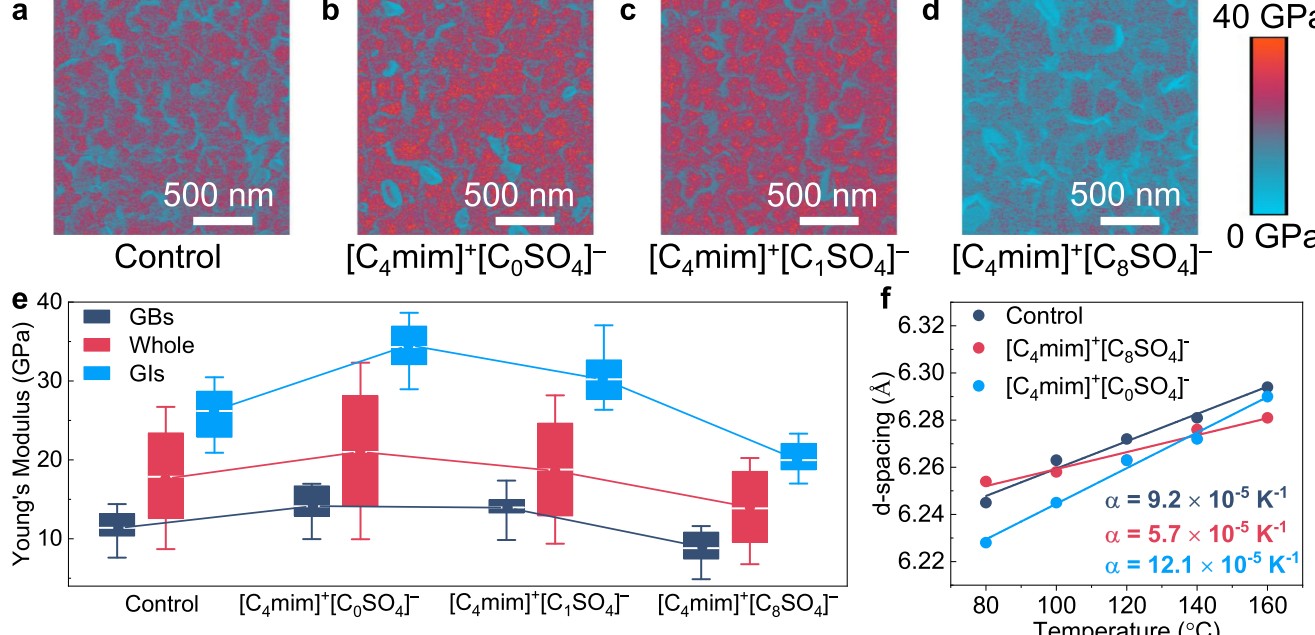

**Fig. 1 | Reduction of residue stress via LASs. a–d** PFQNM-AFM mapping of perovskite thin films. Young's Modulus maps of perovskite film without (referred as control sample) (**a**) additives, $[C_4mim]^+[C_0SO_4]^-$ (**b**), $[C_4mim]^+[C_1SO_4]^-$ (**c**) and $[C_4mim]^+[C_8SO_4]^-$ (**d**) treated films. The colored scale bar represents the magnitude of modules. **e** The box plot of the YM values across the whole images (red) and separately for the regions identified as GBs (navy) or as GIs (blue). The box plot denotes the median (center line), 75th (top edge of the box) and 25th (bottom edge of the box) percentiles. The colored curves are the connecting line corresponding to the mean of the statistical data points. **f** Temperature-dependent (001) d–spacing of control, $[C_4mim]^+[C_0SO_4]^-$ and $[C_4mim]^+[C_8SO_4]^-$ treated films.

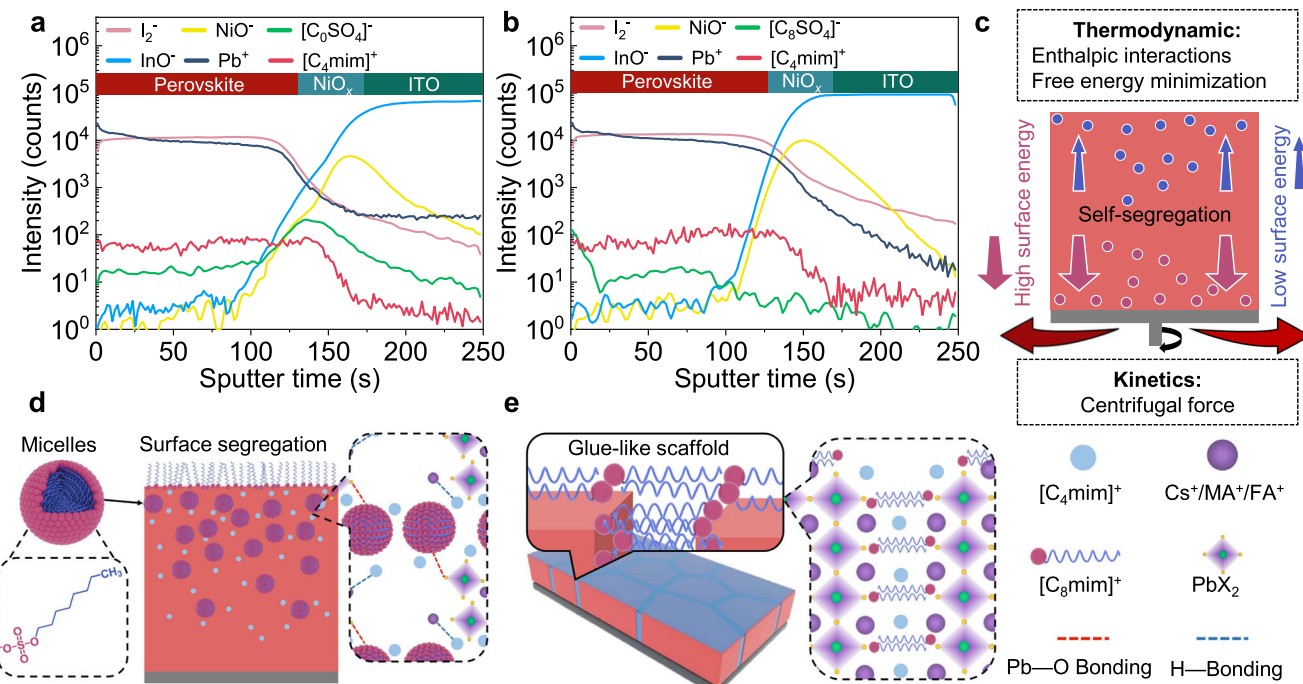

**Fig. 2 | Segregation and micellization of LASs. a, b** ToF-SIMS depth profiles of the [C$_4$mim]$^+$[C$_0$SO$_4$]$^-$ (**a**) and [C$_4$mim]$^+$[C$_8$SO$_4$]$^-$ (**b**) treated perovskite films. Both samples display the uniform Pb$^+$ and I$_2^-$ distributions throughout the entire perovskite bulk, in good agreement with the component distributions acquired from the control sample without additive (Supplementary Fig. 7). The existence of NiO$^-$ and InO$^-$ ions can be associated with the spatial distribution of NiO$_x$, ITO, respectively. **c** The schematic for the mechanism of additive segregation during the film formation. **d, e** The schematic for the spatial distributions of [C$_4$mim]$^+$[C$_8$SO$_4$]$^-$ additives in perovskite precursor and the interaction between [C$_4$mim]$^+$[C$_0$SO$_4$]$^-$ and perovskite precursors (**d**). The role of the additive segregation in perovskite film formation and mechanical engineering (**e**).

comparison since it has been demonstrated in recent reports to improve the efficiency and stability when added into perovskite precursors[30]. On the contrary, the grains became larger (Supplementary Fig. 2a, c) for the [C$_1$SO$_4$]$^-$–based perovskite (Fig. 1c), accompanied by the decreased YMs in both GI and GB regions. The trend is further enhanced (Supplementary Fig. 2a, d) for the perovskite with eight-alkyl-chain [C$_4$mim]$^+$[C$_8$SO$_4$]$^-$ (Fig. 1d), showing a significantly reduced average YM of 14.1 GPa. This suggests the softening effect from the flexible long alkyl chains, which are mainly located near the GBs as evidenced by the extended low-YM area. Since the intermolecular van der Waals (VDW) interactions become stronger with increasing alkyl-chain length[16], the long alkyl chains in [C$_4$mim]$^+$[C$_8$SO$_4$]$^-$ form a soft framework around the perovskite grains[29], and act as a "plasticizer"(ref. 25) to reduce the YM of the perovskite film (see details below). These observations are further highlighted in Supplementary Fig. 4, where topography and YM maps are overlaid.

We then performed the temperature-dependent X-ray diffraction (XRD) (Supplementary Fig. 5) to explore the effect of [C$_4$mim]$^+$[C$_8$SO$_4$]$^-$ additives on the CTE and residual stress of perovskite films (Supplementary Note 1). For the control sample, the scattering peaks (2$\theta$) gradually shifted to lower angles with the increased annealing temperature from 14.17 to 14.09°, indicating the increase of crystal plane distance d(001). The CTE is extracted to be $9.2 \times 10^{-5}$ K$^{-1}$ (Fig. 1f), close to the previously reported values for MAPbI$_3$ (ref. 13), and the calculated stress is as high as 148.5 MPa (at the annealing temperature of 100 °C), confirming that the control film suffers from a serious tensile stress due to its soft lattices[11]. Similarly, the [C$_0$SO$_4$]$^-$ – based film exhibits an even larger CTE of $12.1 \times 10^{-5}$ K$^{-1}$ with a higher stress of 232.8 MPa. In contrast, the [C$_8$SO$_4$]$^-$ – based perovskite delivers a significantly reduced shift of diffraction peaks, exhibiting a low CTE of $5.7 \times 10^{-5}$ K$^{-1}$ and thus a remarkably reduced stress of only 65.3 MPa. This implies that residual tensile stress in the perovskite films can be significantly suppressed by the additive of long-alkyl-chain

[C$_4$mim]$^+$[C$_8$SO$_4$]$^-$. The results can also be confirmed by the grazing-incidence X-ray diffraction (GIXRD) (Supplementary Fig. 6 and Note 2) and Raman spectroscopy (Supplementary Fig. 7 and Note 3) measurements, respectively.

## Segregation and micellization of LASs

The above-mentioned variations in grain sizes and film stresses inspire us to get further insights into the spatial distributions of [C$_4$mim]$^+$[C$_0$SO$_4$]$^-$ (Fig. 2a) and [C$_4$mim]$^+$[C$_8$SO$_4$]$^-$ (Fig. 2b) in perovskite films via the time-of-flight secondary-ion mass spectrometry (ToF-SIMS) technique. Compared with the control sample without additive (Supplementary Fig. 8), the presence of [C$_4$mim]$^+$ signal in the [C$_4$mim]$^+$[C$_0$SO$_4$]$^-$ treated sample suggested that the [C$_4$mim]$^+$ cations were uniformly distributed throughout the film, similar to the additive of [C$_4$mim]$^+$[BF$_4$]$^-$ as previously reported[31], while the [C$_0$SO$_4$]$^-$ anions were gradient distributed in the bulk perovskite, mainly enriched at the buried interface, i.e., the downward segregation of [C$_0$SO$_4$]$^-$. In stark contrast, for the [C$_4$mim]$^+$[C$_8$SO$_4$]$^-$ treated sample, the [C$_8$SO$_4$]$^-$ ion signal increased by one order of magnitude at the surface, indicating the accumulation of [C$_8$SO$_4$]$^-$ ions at the top surface of perovskite and the downward gradient distribution in the bulk film, i.e., the upward segregation of [C$_8$SO$_4$]$^-$. This distribution difference can be further visualized in the three-dimensional (3D) elemental maps (Supplementary Fig. 9). It should be noted that the radius of both anions and cations (~324, ~224 and ~357 pm for [C$_4$mim]$^+$, [C$_0$SO$_4$]$^-$ and [C$_8$SO$_4$]$^-$, respectively) are so large that they tend to locate at the perovskite GBs rather than to substitute into the crystal lattice, consistent with the XRD results (Supplementary Fig. 10).

To date, both the upward and downward segregation of the additives in perovskite films has been reported by numerous groups[30,32,33], however, the exact mechanism is still unclear. At this point, we declare that the common denominator of both segregation mechanisms is the system's free energy reduction[34]. During the spin

coating of the precursor solution (Fig. 2c), the centrifugal force together with the unfavorable enthalpic interactions between additive and perovskite components provides a strong driving force for the additive segregation[35]. Thermodynamic considerations dictate that the high-surface-energy polar $[C_0SO_4]^-$ moiety tends to migrate toward the substrate, i.e., downward segregation, to minimize the system's free energy[36]. Conversely, the low-surface-energy $[C_8SO_4]^-$ moiety with non-polar long-alkyl-chain backbones prefers to enrich the top surface to reduce the overall free energy of the system by substituting for high-surface-energy perovskite components at the surface, i.e., upward segregation. Therefore, these $[C_8SO_4]^-$ moieties spontaneously migrate to the solution-air interface and self-limited to a monolayer with the hydrophobic tails pointing to air and the hydrophilic heads dissolved in the solution region.

We then discuss below the role of the additive segregation in the perovskite nucleation and mechanical engineering. Different from the control (Supplementary Fig. 11) and $[C_0SO_4]^-$ (Supplementary Fig. 12) based samples (Supplementary Note 4), the $[C_4mim]^+[C_8SO_4]^-$, as a classical LAS which contains eight carbon atoms on the anion alkyl chain, has been reported to undergo micellization in various solutions when exceeding its critical micelle concentration. Since the additive concentration can be simply boosted by the anti-solvent dripping, the amphiphilic $[C_8SO_4]^-$ anions can spontaneously form the micellar-like aggregates near the surface of the wet film, with the hydrophobic (nonpolar) tails toward the center and the hydrophilic (polar) head groups at the outer surface in a polar environment (here the dimethylformamide and dimethyl sulfoxide) (Fig. 2d). In parallel, the $[C_4mim]^+$ cations are adsorbed as counterions in the vicinity of the micellar surface to effectively neutralize the charged head groups by screening the intra-micellar electrostatic repulsion among the polar head groups, further facilitating the micelle formation[37]. Due to the strong electrostatic attraction between the S=O and $Pb^{2+}$ (Supplementary Fig. 13), the micelles can link small perovskite colloids to form the large pre-nucleation clusters (Supplementary Fig. 14). According to the non-classical nucleation theory, these pre-nucleation clusters can accelerate the nucleation process and serve as preferential nucleation centers by lowering of the Gibbs free energy barrier for heterogeneous nucleation[38], subsequently acting as seeds to guide the further perovskite crystallization (Supplementary Fig. 15).

In the following growth processes, the $[C_8SO_4]^-$ anions are automatically expelled to the perovskite surface and GBs, with the S=O head groups anchoring to undercoordinated $Pb^{2+}$ (Supplementary Fig. 16) and the hydrophobic alkyl chain tails gathering around the perovskite grains (Fig. 2e). Specifically, the highly polar S=O groups tend to adhere to more charged crystal faces and slow those growth directions[39], beneficial for achieving a highly crystallized film with preferential orientation along the neutral (001) plane (Supplementary Fig. 17 and Note 5). Meanwhile, the relatively small-sized $[C_4mim]^+$ cations, which are homogeneously distributed at the periphery of the perovskite grains, act as a "reservoir" to retard the intercalation of $FA^+$ into Pb–I frameworks through the hydrogen-bonding between the H atom of the amidine in FA and the N3 atom of the imidazole ring in $[C_4mim]^+$ (Supplementary Fig. 18)[40]. The nonpolar monolayer of long-alkyl-chain backbones, which is spontaneously self-limited on the solution surface as mentioned above, plays a "solvent molecular sieve" role in alleviating the evaporation of polar solvent[41]. The $SO_4^{2-}$, as a strong kosmotropic ion, can foster the strongly H-bonded network in polar solvent molecules, also slowing down the evaporation[42]. These three effects synergistically slow down the crystal growth and further enlarge the grain size. The enlarged perovskite grain size can also be confirmed by the scanning electron microscopy (SEM) results (Supplementary Fig. 19). After film formation, a soft and robust supporting scaffold embracing the perovskite grains is formed by the $[C_8SO_4]^-$ anions, which can strongly interact with neighboring grains via coordination reactions and physically link them together through the VDW

interactions between long alkyl chains, forcing the grains to contact with each other more tightly and maintaining the film integrity under stresses (Fig. 2e).

It is well known that the intrinsic elastic properties of the perovskite with the structure of ABX3 are determined by (1) the B–X bond strengths within the $BX_6^{4-}$ framework and (2) the interaction between A cation and $BX_6^{4-}$, and (3) the relative packing density within perovskite[43]. Correspondingly, since the VDW force between alkyl chains is an order of magnitude less than the electrostatic force presents within the B–X framework[44], and the hydrogen-bonding between A cations and B–X framework can be weakened by the $[C_4mim]^+$ as mentioned above, the distribution of $[C_4mim]^+[C_8SO_4]^-$ in the overall structure can significantly reduce the stiffness, i.e., YM, of the perovskite grains, beneficial for the efficient release of applied stresses[45]. At first sight, the enlarged grain sizes here can readily improve the mechanical stability of the perovskite films by reducing the defect-laden GBs, which are the weakest mechanical links in polycrystalline films[46,47]. In addition, the long-alkyl-chain scaffold, with a much lower stiffness relative to perovskite grains, can further function as a stress absorber to maintain stronger intermolecular interactions at larger stretching distance when the stresses are applied to the perovskite films[48], increasing the crack deflection around GBs and thus promoting the plastic energy dissipation process during cohesive fracture[16]. More importantly, the tightly adhered scaffold also plays a role of surface lattice "tape" to hinder the lattice expansion or contraction, i.e., reduce the perovskite CTE, and to significantly buffer the residual stresses induced by the volume variation. Moreover, this hydrophobic scaffold further creates a moisture-repelling barrier for isolating the perovskite grains from environment, improving the stability of the perovskite. Therefore, we conclude that the long-alkyl-chain anionic surfactant additive $[C_4mim]^+[C_8SO_4]^-$ can not only chemically ameliorate the perovskite crystallization kinetics based on the synergistic effects of the surface segregation, micellar-like self-organization and pre-nucleation clusters, but also provide a glue-like scaffold to physically embrace the perovskite grains for improving the perovskite mechanical properties and eliminating the film residual stresses.

## Properties of LAS-modified perovskite films

Stress engineering has been reported as an effective way to reduce defects[49], suppress ion migration[50], improve energy level alignment[10] and facilitate the lattice ordering[51] of perovskite films, thereby ultimately increasing the device performance and improving the device stability[9]. In this section, we first investigate the effect of LASs on the defect density in perovskite films. The steady-state photoluminescence (SSPL) and time-resolved PL (TRPL) measurements were performed for the perovskite films without (control) and with (target) LASs. Both perovskite films have similar light absorbance at the excitation wavelength (Fig. 3a). After the LASs modification, the emission intensity of the perovskite film is significantly enhanced, and the emission peak is bule shifted to a short wavelength in a certain extent. Correspondingly, the carrier lifetime is elongated from 238 ns to 873 ns (Fig. 3b), indicating the reduction of nonradiative recombination. This can also be confirmed by the results of the photoluminescence quantum yield (PLQY) measurements[52] (Supplementary Fig. 20). Pump fluence dependent PL measurements were then applied to quantitatively determine the concentration of trap states in perovskite films[53] (Fig. 3c). The calculated surface and bulk trap densities are reduced from $3.57 \times 10^{17}$ and $0.23 \times 10^{17}$ cm$^{-3}$ for control samples, to $2.17 \times 10^{17}$ and $0.14 \times 10^{17}$ cm$^{-3}$ for target ones (Supplementary Note 6 and Table 3), respectively, in good agreement with previously reported values[54] and confirming the minimized trap density.

We then performed conductive atomic force microscopy (c-AFM) measurements to study the conductivity difference between the GIs and the GBs induced by the LASs (Fig. 3d, e and Supplementary Fig. 21), since the GBs dominate the ion migration pathway in polycrystalline

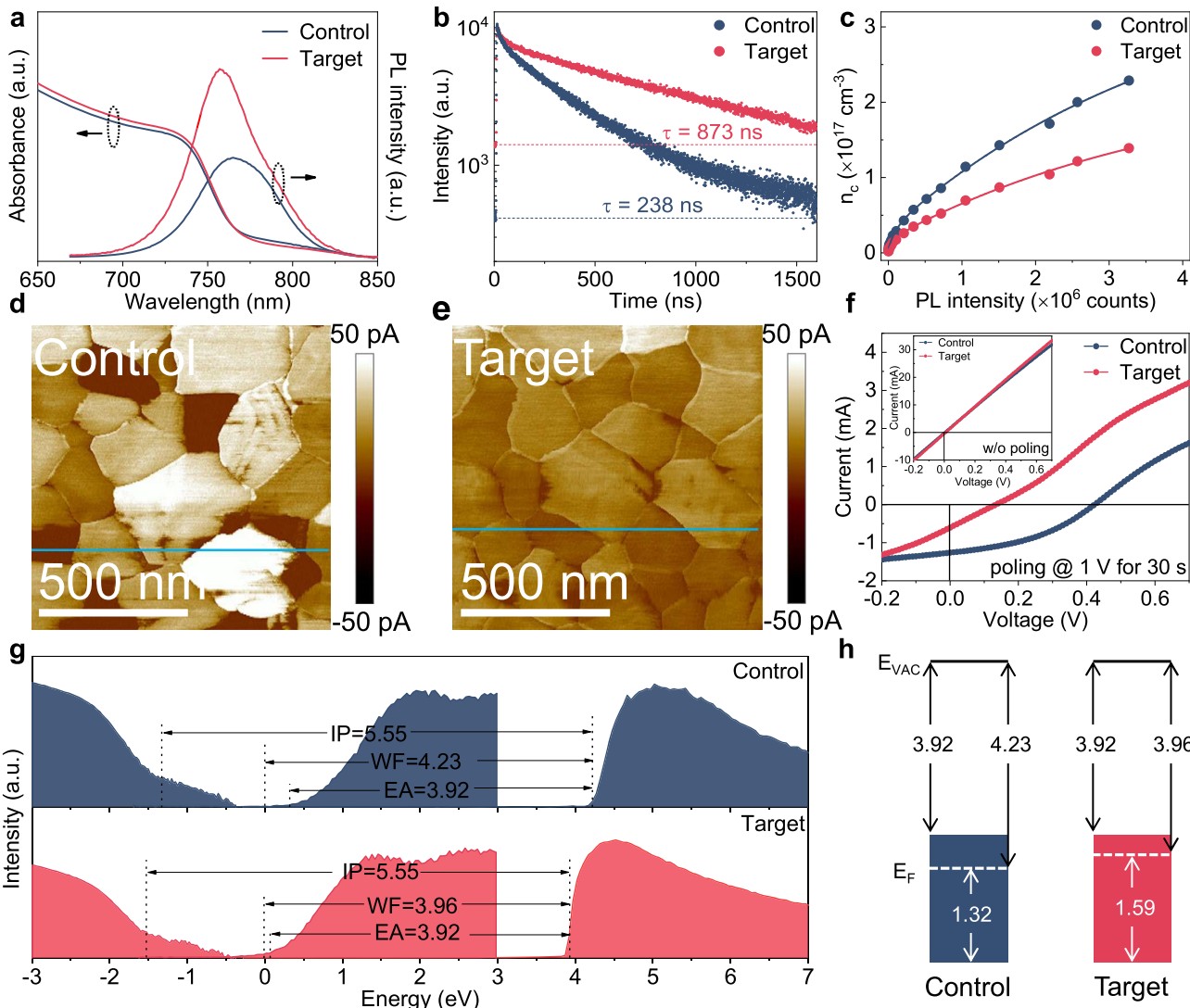

**Fig. 3 | Properties of LAS-modified perovskite films. a** UV-vis absorption and SSPL spectra of perovskite film without additives (referred as control) and [C₄mim]⁺[C₈SO₄]⁻ treated perovskite films (referred as target). **b** TRPL decay curves of control and target films. The dashed lines indicate the background levels. **c** Pump fluence dependent PL measurements of control and target films. The solid lines represent the best fits obtained with equation (3) in Supplementary Note 6. **d**, **e** The c-AFM images of control (**d**) and target (**e**) films. **f** The current-voltage curves of the control and target devices after and before (inset) poling. **g** UPS and IPES of secondary electron cutoff region, valence and conduction band regions of control and target perovskite films. **h** Energy levels derived from UPS and IPES spectra.

films. The current flowing through the perovskite grains is reduced from ~22 pA for control to ~10 pA for target samples, and no significant conductivity contrast between the GBs and GIs can be observed for the target perovskite surface (Supplementary Fig. 21), strongly demonstrate the suppression of ion migration induced by the LASs modification. We then examined the ion migration behavior of perovskite films through electrical poling experiments with the structure of tin-doped indium oxide (ITO)/Au/perovskite/Au (Supplementary Fig. 22 and Note 7). For both samples, the unpolarized devices yield zero open-circuit voltage ($V_{OC}$) under 1 sun illumination due to the symmetry of the electrode structure (Fig. 3f). While after poling the device at 1 V for 30 s, the target device exhibits a much lower $V_{OC}$ of 0.14 V, which is decreased by 67% in comparison with that of the control sample (0.42 V) (Fig. 3f), indicating the reduced ion migration in the target sample. To further verify this, a temperature-dependent conductivity measurement was performed to obtain the activation energy (Ea) for ion migration. Based on the Arrhenius plot shown in Supplementary Fig. 23, we can extract an Ea of 0.142 eV for control films, which is comparable with the value in previously reported perovskites[55]. By contrast, we observe a nearly double Ea (0.252 eV) in

target films, proving that halide migration has been substantially suppressed in the target films. This result can also be confirmed by the transient ionic current (TIC) measurements[56] (Supplementary Fig. 24) as the reduced mobile ion concentrations in the target samples ($3.08 \times 10^{18}$ cm⁻³) compared with control samples ($9.26 \times 10^{18}$ cm⁻³).

Moreover, the impact of LASs on the perovskite electronic structure was also investigated by Ultraviolet photon spectroscopy (UPS) and inverse photoelectron spectroscopy (IPES) measurements (Fig. 3g). The Fermi level ($E_F$) is up-shifted by 270 meV toward the conduction band minima (CBM) in the target perovskite relative to that of the control one, indicating the n-type doping of perovskite surface energetics (Fig. 3h). This behavior is consistent with the results obtained from the Kelvin probe force microscopy (KPFM) (Supplementary Fig. 25) and Hall effect measurements, where the electron concentration and mobility are increased from $4.1 \times 10^{17}$ cm⁻³ and 0.97 cm²V⁻¹s⁻¹ for control to $1.5 \times 10^{18}$ cm⁻³ and 2.9 cm²V⁻¹s⁻¹ for target samples. It is well known that the $E_F$ shift in the perovskite film originated from a change in the ratio of lead halide- to organic halide-terminated surfaces[57], and the energy level of the organic iodide termination is shallower than that of the lead iodide termination[58]. Here,

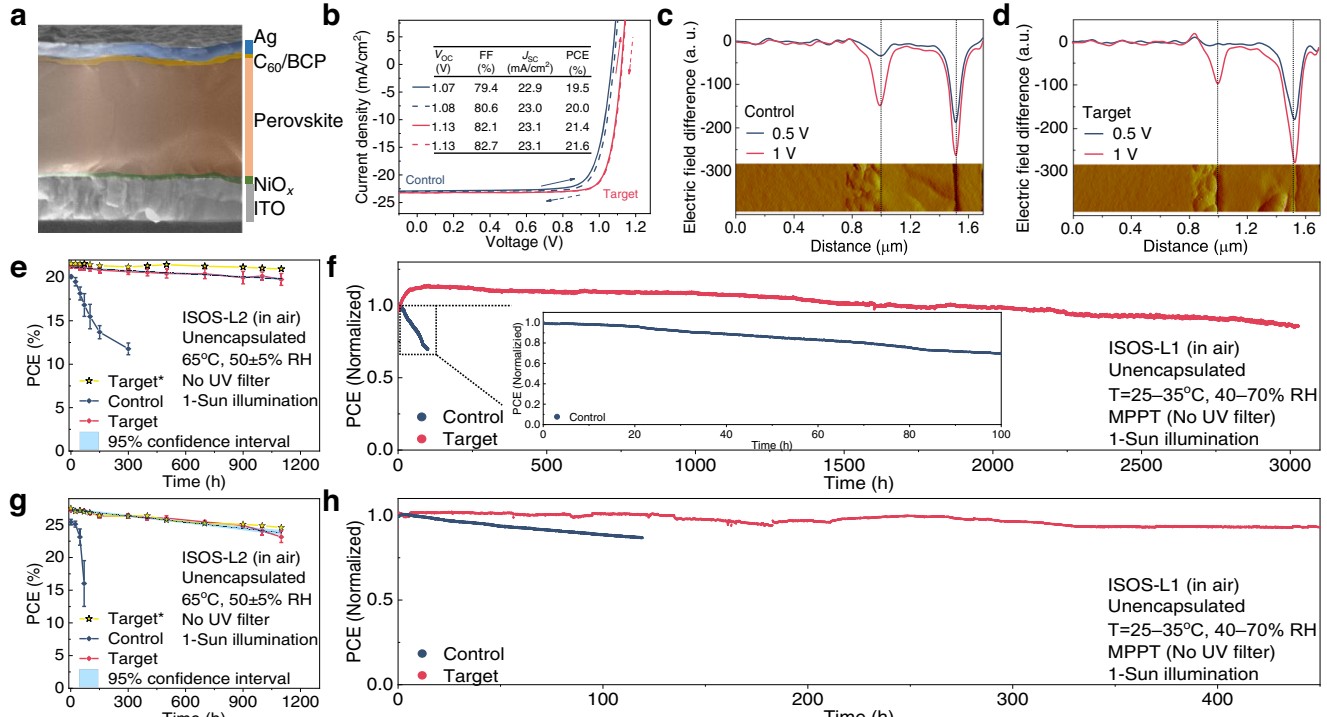

**Fig. 4 | Device performance and stability. a** Cross-sectional SEM image of the single-junction PSC. **b** J–V curves of the best-performing control (navy) and target (red) devices, in forward scan (solid lines) and reverse scan (dotted lines). The inset table shows the corresponding efficiency parameters. **c**, **d** AFM cross-section images (inset) and electric field differences across the whole control (**c**) device and target (**d**) device, taken by the first derivative in potential differences with respect to the zero bias. **e** ISOS-L2 measurements of unencapsulated devices. The target champion cell (denoted with*) is indicated with yellow stars (surrounded by black border lines). Standard deviation (error bar) is calculated from six individual devices in the same batch. The 95% confidence interval for the stability of the target devices is shown as a blue band. **f** The operational stability of the unencapsulated control and target single-junction devices under MPPT in air. **g** ISOS-L2 measurements of unencapsulated tandem devices. The target champion tandem cell (denoted with*) is indicated with yellow stars (surrounded by black border lines). Standard deviation is calculated from six individual devices in the same batch. The 95% confidence interval for the stability of the target tandem devices is shown as a blue band. **h** MPPT of the unencapsulated control and target tandem devices in air.

for the target perovskite with the $[C_4mim]^+[C_8SO_4]^-$ additive, the anions $[C_8SO_4]^-$ mainly distributed at the top of perovskite film (Supplementary Fig. 9) can bind to lead iodide termination (Supplementary Fig. 16). Therefore, the surface is dominated by the organic halide, indicating a more n-type nature for the target film. Meanwhile, the target film exhibits an upshift in the valence band maxima (VBM) value relative to the control one, whereas the CBM value is constant within the experimental error of ±0.1 eV. This upshift of the VBM and thus the narrowing of the perovskite surface bandgap is likely attributed to the shortened Pb–I bonds and/or the increased Pb–I–Pb angles under the reduced tensile stress, which could increase the antibonding overlap between the Pb 6 s and I 5p orbitals[59]. Here, the UPS and IPES measurements are highly dependent on the surface properties, thus the extracted electronic bandgap is larger than the optical (bulk) bandgap attained from the Tauc plots (Supplementary Fig. 26), in good agreement with the previous reports[60]. These results imply that the stress reduction can upshift both the $E_F$ and VBM at the perovskite surface, leading to a more downward energy band bending as well as a larger VBM offset at the interface between perovskite and $C_{60}$ electron transporting layer (ETL) (Supplementary Fig. 27) and thus causing a strong built-in electric field and a high hole transport barrier to be more favorable for enhancing the electron selectivity[61].

## Device performance and stability

To explore the influence of the stress engineering on the photovoltaic performance, *p-i-n*–type single-junction perovskite solar cells (PSCs) with the architecture of ITO/NiO$_x$/Cs$_{0.05}$(FA$_{0.83}$MA$_{0.17}$)$_{0.95}$Pb(I$_{0.82}$Br$_{0.18}$)$_3$/C$_{60}$/BCP/Ag were fabricated. The corresponding device configuration

and cross-sectional SEM image are shown in Supplementary Fig. 28 and Fig. 4a, respectively. By optimizing the concentration of $[C_4mim]^+$ $[C_8SO_4]^-$, the target device achieves a champion PCE of 21.6% under the reverse scan, far surpassing the best control one (Supplementary Fig. 29). The current density–voltage (J–V) and stabilized power output (SPO) curves of the best-performing control and target devices are shown in Fig. 4b and Supplementary Fig. 30, respectively. Negligible hysteresis can be observed for both devices. The current density ($J_{SC}$) calculated from external quantum efficiency (EQE) measurements (Supplementary Fig. 31) agrees with the measured $J_{SC}$ from J to V scanning (<5% error). The photovoltaic parameters based on 12 devices suggest the good reproducibility of the target device (Supplementary Fig. 32). The performance of our device is comparable to the state-of-the-art device with the bandgap of 1.63 eV (Supplementary Table 4). To evaluate the effects of the surfactant additives consisting of $[C_nSO_4]^-$ featuring different alkyl chain lengths as well as $[C_4mim]^+[BF_4]^-$ at the device level, we also performed J–V and SPO curves of the surfactant additives treated devices as shown in Supplementary Fig. 33.

The cross-sectional KPFM measurements were further performed to elucidate the causes of the improved device performance (Supplementary Fig. 34). Compared with the control device (Fig. 4c), the target device exhibits a stronger electric-field difference peak at the ETL side, but a weaker peak at the HTL side (Fig. 4d and Supplementary Fig. 35). This increased voltage drop between two interfaces is beneficial to decrease the energy loss for the majority-carrier transport, thus contributing to an improved $V_{OC}$ and fill factor (FF)[62]. The defect density was then quantitatively evaluated by the space-charge-limited current (SCLC) method in both hole-only (with the structure of ITO/NiO$_x$/

perovskite/Spiro-OMeTAD/Ag) and electron-only (with the structure of ITO/SnO$_x$/perovskite/C$_{60}$/BCP/Ag) devices (Supplementary Fig. 36). As expected, for both hole- and electron-only structures, the target devices exhibit smaller $V_{TFL}$ and lower electron trap-state densities than the control ones. This can be also confirmed by plotting $V_{OC}$ as a function of light intensity (Supplementary Fig. 37), where the ideal factor was minimized from 1.52 $k_BT$/q for the control device to 1.26 $k_BT$/q for the target device.

Time-dependent XRD measurements were performed to investigate the effect of LASs on the light stability of perovskite films. After a 14-day illumination under a xenon lamp with equivalent 1 sun AM 1.5 G in the nitrogen atmosphere (Supplementary Fig. 38), the target sample showed a smaller ratio of the peak intensity of PbI$_2$ to perovskite than the control one in the XRD spectra, indicating the improved light stability induced by LASs. However, it is well known that both defect passivation and stress reduction can synergistically contribute to the perovskite stability[9,63]. To distinguish between them, three typical samples with different stresses, i.e., pristine (without passivation and stress control), [C$_0$SO$_4$]$^-$–based (with passivation but increased stresses) and [C$_8$SO$_4$]$^-$–based (with passivation and decreased stresses) perovskite films, were chosen for further comparison. All samples were measured by XRD every 7 days under continuous illumination. For the pristine perovskite, the peak intensity ratio of PbI$_2$ to (001) perovskite is significantly increased after continuous illumination, indicating the serious perovskite decomposition. Meanwhile, the perovskite peak gradually shift from 14.13 to 14.07° during the photoaging (Supplementary Fig. 39a, d), demonstrating that the illumination-induced perovskite decomposition is accompanied by a strong stress relaxation, consistent with the results reported by Zhao[13]. A similar trend can also be observed in the [C$_0$SO$_4$]$^-$ – based films, but the peak intensity ratio of PbI$_2$ to perovskite is reduced and the shift of the perovskite peak is enlarged, implying the slightly suppressed perovskite decomposition and the increased stress relaxation (Supplementary Fig. 39b, e). This indicates that the stability contribution from the defect passivation of [C$_4$mim]$^+$[C$_0$SO$_4$]$^-$ is compromised by the increased stresses, thus resulting in a limited stability improvement. In contrast, for the [C$_8$SO$_4$]$^-$ based samples, no obvious PbI$_2$ peak can be observed, and the perovskite peak of perovskite is firmly located at a lower angle of 14.14° (Supplementary Fig. 39c, f), suggesting the significantly suppressed illumination-induced perovskite decomposition, i.e., the excellent light stability, due to the synergistic effects of defect passivation and stress reduction. This phenomenon can also be confirmed by the TRPL measurements for aged samples (Supplementary Fig. 40), where the less stressed target perovskite maintains a high lifetime of 716 ns. However, the lifetime of the highly stressed perovskite based on [C$_0$SO$_4$]$^-$ sharply decreased from 762 ns to 142 ns, even comparable to the value of 128 ns for the pristine sample without any passivation.

Apart from illumination, the moisture and the heat also have great effects on the stability of the perovskite. The thermal stability was first examined by holding the perovskite films at 150 °C for 20 min in the ambient atmosphere before collecting XRD scans (Supplementary Fig. 41). An obvious PbI$_2$ diffraction peak appears for the control perovskite, while only a slight PbI$_2$ peak can be observed for the target film. The static contact angle (CA) of water on the perovskite surface was reported to be correlated with the moisture stability of the perovskite. The target film exhibits an enhanced hydrophobicity (with CA of 52.3°) over the control one (with CA of 27.5°), suggesting the better moisture tolerance (Supplementary Fig. 42a, b). Indeed, when immersing in water for only 5 s, the control perovskite is completely decomposed to a yellow PbI$_2$ while the target film still maintains the black appearance (Supplementary Fig. 42c, d). We then monitored the shelf-stability of the devices at 85 °C and 85% relative humidity (RH) under illumination with the intensity of 100 mW/cm$^2$. The moisture quickly penetrated into the perovskite layer and decomposed it into yellow PbI$_2$ in just 10 min for the control device, while no obvious

yellow PbI$_2$ can be observed for the target device even after 30 min aging (Supplementary Fig. 43). The only variation is the degradation of the Ag electrodes by the corrosive gas emanating from the decomposed perovskite, consistent with the results reported by Hatton et al.[64].

We evaluate the device stability under accelerated-aging conditions according to International Summit on Organic Photovoltaic Stability (ISOS) protocols. The unencapsulated devices were monitored on a temperature-controllable stage (65 °C) for 1100 h (ISOS-L2)[65], and the results are shown in Fig. 4e and Supplementary Fig. 44. The average PCEs of the six target devices retain 92.8% of the initial values, with the best device retaining 97.1% of initial PCE for 1100 h, greatly surpassing the control ones. We also presented the J–V and the measured SPO curves at different aging times in Supplementary Fig. 45, clearly showing the evolution of device performance (including the increased hysteresis in the J–V curves) during ISOS-L2 aging. Besides, we observed a noticeably slow increase of hysteresis in the target device according to the recorded J–V curves. Moreover, the SPO values also degrade more quickly than the J–V-determined efficiency, and we observed roughly 12 and 9% SPO decrease in the initial performance after 1100 h of aging (Supplementary Fig. 45) in the target cells and the champion cell, respectively. By comparison, a roughly 50% drop in the SPO over the 300 h of aging was observed for the control cells. The operational stability of the unencapsulated devices (Fig. 4f) was further monitored under maximum power point tracking (MPPT) under the continuous illumination system equipped with a xenon lamp (including UV irradiation) in air without any temperature and humidity control (with the temperature and humidity during the test shown in Supplementary Fig. 46). The control device showed severely deteriorated operational stability after merely 100 h. By contrast, the target device showed only less than a 15% efficiency drop after 3000 h, being the best MPPT stability result for PSCs reported so far under similar test conditions (Supplementary Table 1). Having demonstrated the improved stability of [C$_4$mim]$^+$[C$_8$SO$_4$]$^-$-containing perovskite devices, we also proceed to investigate the MPPT stabilities of the complete photovoltaic cells incorporated with different surfactant additives as shown in Supplementary Fig. 47.

Finally, we fabricated monolithic perovskite/silicon tandem devices, employing a silicon bottom cell with a double-side tunneling oxide passivating contact (TOPCon) structure, i.e., featuring the configuration of Ag/Cr/poly-Si(p+)/silicon oxide (SiO$_x$)/n-Si/SiO$_x$/poly-Si(n+)/indium zinc oxide (IZO)/NiO$_x$/perovskite/C$_{60}$/BCP:Ag/LiF$_x$/IZO/Ag and device cross-section SEM image in Supplementary Fig. 48. The J–V (Supplementary Fig. 49) and EQE (Supplementary Fig. 50) curves of champion tandem devices with corresponding performance parameters are summarized. The champion target device shows a notable PCE of 27.4% (27.3%) with the FF of 81.8% (81.5%), a $V_{OC}$ of 1.78 V (1.78 V) and a $J_{SC}$ of 18.8 mA cm$^{-2}$ (18.8 mA cm$^{-2}$) under the reverse (forward) voltage scan. We also obtained the stabilized PCE of 27.3% by tracking the MPP of the tandem device for 10 min (Supplementary Fig. 51), in good agreement with the J–V results (Supplementary Fig. 52). To verify our in-house measurement, one of the devices was sent to Shanghai Institute of Microsystem and Information Technology (SIMIT) for the PV certification using dual-source, achieving a steady-state PCE of 26.3% (Supplementary Fig. 53). Inspiringly, in contrast to the unstable control tandem device, the target one (unencapsulated, in air, 20−35 °C, 40−70% RH, using xenon-lamp) exhibits an excellent stability, retaining 85.0% and 93.6% of their initial PCE values under ISOS-L2 conditions for 1100 h (Fig. 4g) and MPPT ($V_{MPP} = 1.56$ V) for 450 h in the operational conditions (Fig. 4h), respectively. Moreover, the target champion tandem cell retains 89.3% of the initial PCEs for 1100 h under ISOS-L2 conditions. The performance statistics (Supplementary Fig. 54) for ISOS-L2 measurements and the specific temperature and humidity details during the MPPT test are shown in Supplementary Fig. 55. To our knowledge, this is one of the best stabilities achieved in

an unencapsulated tandem device at the MPPT under similar test situation (Supplementary Table 2). Of note, the stability of the perovskite/silicon tandems is still much worse than the single-junction perovskite devices. This seems to be a common problem for the substrate-configuration type perovskite tandems, where the free carries accumulated at the poor ETL/perovskite interface (Supplementary Fig. 56) will reduce the ion migration activation energy and then accelerate the perovskite degradation. Similarly, since the perovskite top cell here is not the limiting cell and thus the photoexcited charge carriers in the perovskite layer are not all extracted, the long-term stability of the perovskite-based tandems could be affected by this current mismatch[66], and may be resolved in the future study.

## Discussion

In summary, we demonstrated that the LAS as an additive to the perovskite precursor solution can not only improve the perovskite crystallization but, more importantly, can also eliminate the film residual stresses. This enables the modified perovskite films with the simultaneously reduced defects, suppressed ion migration and improved energy level alignment. The resultant unencapsulated single-junction perovskite and dual-junction perovskite/silicon tandem devices, with the PCEs of 21.6%, and 27.4%, retain 85.7% and 93.6% of their performance after 3000 h and 450 h of continuous illumination under MPPT operating conditions in air, respectively, representing one of the highest lifetimes reported to date under similar conditions. We believe that the LAS strategy adopted here will pave the way toward commercial production of efficient and stable perovskite-based devices in the near future.

## Methods

### Materials

All materials were used as received without further purification. Methylammonium bromide (MABr) (GreatCell Solar Materials), formamidinium iodide (FAI) (GreatCell Solar Materials), cesium iodide (CsI) (Sigma-Aldrich), lead (II) bromide (PbBr$_2$) (TCI) and lead (II) iodide (PbI$_2$) (TCI) powders were used to make perovskite precursors. 1-Butyl-3-methylimidazolium hydrogen sulfate ([C$_4$mim]$^+$[C$_0$SO$_4$]$^-$), 1-Butyl-3-methylimidazolium methyl sulfate ([C$_4$mim]$^+$[C$_1$SO$_4$]$^-$), and 1-Butyl-3-methylimidazolium octyl sulfate ([C$_4$mim]$^+$[C$_8$SO$_4$]$^-$) were purchased from Sigma-Aldrich. C$_{60}$ (99.5%) and bathocuproine (BCP) (99%) were purchased from Taiwan Lumtec Corp. The anhydrous solvents N,N-dimethylformamide (DMF), dimethyl sulfoxide (DMSO) and chlorobenzene (CB) were purchased from Sigma-Aldrich. The NiO$_x$ nanoparticles and pre-patterned ITO glass were purchased from Advanced Election Technology Co., Ltd. The mask used for depositing metal electrodes was custom-made by Shenzhen Rigorous Technology Co., Ltd.

### Fabrication of the single-junction perovskite solar cells

After cleaning through sequential washing with detergent, deionized water, acetone and isopropanol, the ITO glass was UV ozone cleaned for 20 min. The NiO$_x$ nanoparticle ink was prepared by dispersing the NiO$_x$ nanoparticles in the deionized water for a concentration of 10 mg ml$^{-1}$, and the ITO substrate was spin-coated with the NiO$_x$ nanoparticle ink at 3000 r.p.m. for 30 s. The precursors for the Cs$_{0.05}$(FA$_{0.83}$MA$_{0.17}$)$_{0.95}$Pb(I$_{0.82}$Br$_{0.18}$)$_3$ perovskites were prepared by dissolving the MABr, FAI, PbI$_2$, PbBr$_2$, and CsI in a mixed solvent (4:1 in volume) of DMF and DMSO, respectively. In parallel, the surfactant was dissolved in mixed solvent (4:1 in volume) of DMF/DMSO with the desired molar concentration. Then, the surfactant-containing precursor solutions were prepared by dissolving the perovskite components in the surfactant-containing DMF/DMSO mixed solvent, and all solutions were filtered (0.45 μm, PTFE) before use. The perovskite layers were fabricated according to our previously reported one-step anti-solvent recipe. For fabrication of the perovskite film, the substrate

was spun at 3500 r.p.m. for 35 s with an initial acceleration of 3500 r.p.m., and then 230 μl chlorobenzene was dripped onto the substrate during the last 24 s of spinning. And the substrates were annealed at 100 °C for 20 min. For the electron-transporting (C$_{60}$), hole blocking (BCP) materials and metal electrode (Ag), 24 nm C$_{60}$, 6 nm BCP and 250 nm Ag were sequentially deposited through shadow masks by thermal evaporator (FE thin film Co., Ltd.) placed in a nitrogen-filled glovebox. For the semi-transparent devices, a highly transparent contact of BCP (6 nm):Ag (1 nm)/IZO (100 nm) is applied as a diffusion barrier at the interface between C$_{60}$/Ag (ref. 67). The IZO top electrode (44 Ω/sq) was sputtered (Sky technology development Co., Ltd.) on the buffer layer with radiofrequency power of 80 W over a ø60 mm target (90% In$_2$O$_3$/10% ZnO, 99.9%, Zhong Nuo Advanced Material Technology Co., Ltd., the source to substrate distance is 12 cm). It is worth noting that the first 10-nm IZO deposition was conducted with a lower power of 30 W to minimize the sputtering damage. The thickness of C$_{60}$, BCP and IZO layers was first calibrated by a spectroscopic ellipsometer. The evaporation rate and thickness of each experiment were monitored by quartz crystal microbalance sensors.

### Fabrication of the monolithic perovskite/silicon tandem solar cells

The TOPCon silicon bottom cells were prepared using the n-type Czochralski (CZ) silicon wafers with a thickness of 270 μm and the resistivity of 1−3 Ω cm. Firstly, the wafers were subjected to the standard rear texturing and Radio Corporation of America (RCA) cleaning procedure. Detailed information can be found in our previous work[67]. The ultrathin SiO$_x$ (-1.7 nm) was then prepared by plasma enhanced chemical vapor deposition (PECVD) on the front/rear side of the Si substrates, followed by a p-doped/n-doped a-Si:H (-30 nm). The 900 °C−960 °C high-temperature co-annealing was then carried out in the furnace for 30 min to crystallize the a-Si:H films. To be noted, the SiO$_x$ was formed by the plasma-assisted nitrous-oxide (N$_2$O) gas oxidation (PANO) method. Silane (SiH$_4$), borane (B$_2$H$_6$), phosphine (PH$_3$), hydrogen (H$_2$) and methane (CH$_4$) were used as the reaction gases in the PECVD system (Model-FE-PECVD-L-2). The metallization of the rear contact was then prepared by the sequential deposition of Cr and Ag (1 and 300 nm, respectively) in the thermal evaporation system. The IZO recombination junction (10 nm, >1000 Ω/sq) was sputtered on the front surface. The wafers were then annealed at 150 °C for 10 min to recover the sputtering damage. For the perovskite top cells, the Si bottom cells were then used after UV-ozone treatment for 15 min. After the spin coating of the NiO$_x$ layer, the Cs$_{0.05}$(FA$_{0.83}$MA$_{0.17}$)$_{0.95}$Pb(I$_{0.82}$Br$_{0.18}$)$_3$ perovskite layers were deposited by the one-step anti-solvent recipe. For surfactant-containing precursor solutions, the preparation of the solutions is the same as for single-junction solar cells. Next, the 24 nm C$_{60}$ and the buffer layer (6 nm BCP:1 nm Ag) were deposited by thermal evaporation. Finally, the IZO top electrode (100 nm) was sputtered on the buffer layer with radiofrequency power of 80 W. And the top Ag frame contacts (250 nm) were evaporated around the active area through a shadow mask using the thermal evaporation system with a rate of 1.0 Å/s. And 120 nm of LiF$_x$ antireflection films were evaporated at a rate of 1 Å/s to complete the devices. The thickness of LiF$_x$ antireflection films was also calibrated by spectroscopic ellipsometer.

### Device characterization

The Current-voltage (J−V) curves were performed under the simulated sunlight irradiation of 1 sun (AM1.5 G, 100 mW cm$^{-2}$) using a solar simulator (EMS−35AAA, Ushio Spax Inc.) based on the Ushio Xe short arc lamp 500. The light source was calibrated with the National Renewable Energy Laboratories (NREL)−KG5 filtered silicon reference cell. The J−V curves were measured with a scanning rate of 100 mV s$^{-1}$ (delay time of 10 ms) from 1.2 V to −0.1 V and then back again (from −0.1 V to 1.2 V) for single perovskite solar cells and from 1.9 V to −0.1 V

and then back again (from −0.1 V to 1.9 V) for tandem devices without light soaking or electric-poling preconditioning. The active area was determined with a 0.124 cm² metal aperture placed in front of the solar cell. EQE was measured by an Enli Technology (Taiwan) and recorded with an Enli Technology EQE measurement system (QE−R), and the light intensity at each wavelength was calibrated with standard single-crystal Si and Ge reference solar cells. The perovskite top cell was measured while saturating the Si bottom cell with continuous biased light from a white light equipped with a long-pass (>800 nm) filter. To maintain short-circuit conditions, a bias voltage of 0.5 V was applied during the measurement. The Si bottom cell was measured while saturating the perovskite top cell with continuous biased light from a white light equipped with a low-pass (<800 nm) filter, and the bias voltage was 1 V. All device characterizations were performed at ambient air without any encapsulation.

## Stability test

The stability test was performed on a custom-built system that allowed simultaneous MPPT, and the initial voltage of MPPT ($V_{MPP}$) was determined by a J–V sweep. All of the devices for the stability test were unencapsulated and aged at the regular air ambient environment (RH = 40–70%, T = 20–35 °C) (Supplementary Fig. 40). The solar cell temperature increased to ~30–40 °C under illumination as no active cooling was implemented at the measurement stage. For the operational stability under MPPT, the unencapsulated devices were aged using a xenon lamp (66475-150XF-R22, Newport) light-soaking chamber under simulated full-spectrum AM1.5 sunlight with 100 mW•cm⁻² irradiance. MPPT was collected every second with neither a UV cut-off filter nor a temperature controller. For the ISOS-L2 aging test, the unencapsulated devices were stored in the aging chamber with ~50 ± 5% relative humidity, and the devices were mounted on a metal plate kept at 65 °C using a standard temperature control unit.

## PeakForce quantitative nanomechanical atomic force microscopy (PFQNM-AFM)

PFQNM-AFM allows to simultaneously acquire high-resolution (10 nm × 10 nm) topography and surface YM maps of the perovskite thin film. An advantage of this approach compared to traditional indentation is that PFQNM-AFM does not plastically deform the sample, and therefore does not induce changes in the perovskite material during measurements. Topography and YM maps were acquired using a Bruker Dimension ICON with one advanced PeakForce-QNM mode and the probe tip with a force constant of about 200 N/m and a curvature radius of about 20 nm, which can detect nanoscale heterogeneity. 2 μm × 2 μm areas were scanned with a scan rate of 0.905 Hz and 256 lines each, and 256 force curves were acquired within each line. The YMs were calculated with the extraction of YM/curve, fitting with linearized DMT model for each pixel using Nanoscope (Bruker).

## Other characterizations

XRD measurements were performed by a Bruker AXS D8 Discover diffractometer with Cu Ka radiation (a wavelength of 1.5418 Å). GIXRD patterns were recorded on smartlab XRD. The steady-state PL was measured by a fluorescence spectrometer (HORIBA, FL3-111). Raman spectra were acquired using the Renishaw inVia Reflex. The light source was the 532 nm CW laser at 100 mW power. The ToF-SIMS analysis was conducted using a PHI TRIFT V nanoTOF (ULVAC-PHI, Japan) system via the dual beam slice-and-view analysis scheme. A pulsed 30 keV Bi⁺ ion beam was used as the analysis beam and a 1 keV Cs⁺ ion beam was used as the sputter beam. The analysis area was 96 μm × 96 μm which was at the center of the sputter crater of 250 μm × 250 μm. FTIR spectrum was taken from a NICOLET 6700 (Thermo, America) spectrometer equipped with a DLATGS detector in transmission mode. Dynamic light scattering (DLS) measurements

were performed by a Zetasizer Nano ZS instrument with a 633-nm He-Ne laser. XPS was performed under an ultra-high vacuum with a monochromatic Al Ka X-ray source (Kratos Axis Ultra Ltd.). GIWAXS measurements were carried out with a Xeuss 2.0 SAXS/WAXS laboratory beamline using a Cu X-ray source (8.05 keV, 1.54 Å) and a Pilatus3R 300 K detector. The incidence angle is 0.3°. The SEM images were captured by a field-emission SEM (Hitachi, S4800) at 4 kV acceleration voltage. ¹H–NMR spectra of the Bu and Ea RTMSs were measured on the AVANCE NEO 600 (Switzerland), using DMSO-d₆ as a locking solvent. SSPL and TRPL spectra were measured using the HORIBA FL3-111 spectrometer with the 532 nm excitation of an optical parametric oscillator laser system. The c-AFM and KPFM measurements were performed using a Bruker Dimension ICON atomic force microscope in an Ar-filled glovebox with H₂O and O₂ concentrations <0.1 ppm. For c-AFM measurements, a Pt–Ir coated tip (nanosensor PPP–EFM) was scanned in contact mode and the tip was virtually grounded. The samples were connected to the AFM stage with an applied bias voltage of 1 V. For KPFM measurements, A PPP–EFM tip was scanned in tapping mode for the cross-sectional potential distribution. The ITO was grounded and various bias voltages were applied to the Ag side. No polishing or ion-milling treatment was introduced to the sample preparation to avoid artefacts. The activation energy (Ea) measurement of ions migration was conducted in a probe station (Cindbest CGO-4) under vacuum (1.0 × 10⁻⁴ Pa), and a semiconductor characterization system (Keysight B1500A) was used for the current measurement. UPS and IPES measurements were carried out using AXIS ULTRA DLD (Kratos) and LEIPS (ULVAC–PHI), respectively. The secondary electron cutoff feature (right peak feature in the panel) was shifted by the incident photon energy (21.22 eV) to show the position of the vacuum level ($E_{VAC}$) relative to the $E_F$ (located at 0 eV), which corresponded to the work function (WF). The VBM region (left peak feature in the panel) displayed the position of the valence band reference to the $E_F$. The combination of the WF and VBM gave the position of ionization potential (IP). The electron affinity (EA) positions were obtained from IPES measurements. The drive-level capacitance profiling (DLCP) was measured by an Agilent E4980A.

## Reporting summary

Further information on research design is available in the Nature Research Reporting Summary linked to this article.

## Data availability

Source data are provided with this paper. All the data supporting the findings of this study are available within this article and its Supplementary Information. Any additional information can be obtained from corresponding authors upon request. Source data are provided with this paper.

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

## Acknowledgements

This work was supported by the National Key Research and Development of China (grant no. 2018YFB1500103) awarded to J.Y., National Natural Science Foundation of China (Grant No. 62204245) awarded to Z.Y., the Science and Technology Development Fund, Macao SAR (File No. FDCT-0044/2020/A1, 0082/2021/A2) and UM's research fund (File No. MYRG2020-00151-IAPME) awarded to G.X.

## Author contributions

Conceptualization: X.W., X.Y., and J.Y. Methodology: X.W., Z.Ying, and J.Z. Investigation: X.W., Z.Ying, X.Y., and J.Y. Visualization: X.W., Z.Ying, X.Y., X.L., Z.Z., C.X., M.W., Y.C., Z.Yang and J.X. Funding acquisition: J.Y., J.Sheng, and X.Y. Project administration: J.Sun and J.Sheng. Supervision: J.Y. and X.Y. Writing—original draft: X.W., Z.Ying, X.Y., and J.Y. Writing—review & editing: X.W., Z.Ying, X.Y., J.Y., G.X., and Y.Z.

## Competing interests

The authors declare no competing interests.
