## [Peer Review File · Nature Communications]

Long-chain anionic surfactants enabling stable perovskite/silicon tandems with greatly suppressed stress corrosionREVIEWER COMMENTS

Reviewer #1 (Remarks to the Author):

The manuscript entitled "Segregation and micellization of long-chain anionic surfactants enabling stable perovskite/silicon tandems with greatly suppressed stress corrosion" with Ref #: "NCOMMS-22-46934", by authors "Xinlong Wang, Zhiqin Ying, Jichun Ye et al." is a research article on the

employment of long-alkyl-chain anionic surfactant (LAS) additives: 1-Butyl-3-methylimidazolium hydrogen sulfate ([C4mim]+[C0SO4]-), 1-Butyl-3-methylimidazolium methyl sulfate ([C4mim]+[C1SO4]-), 1-Butyl-3-methylimidazolium octyl sulfate ([C4mim]+[C8SO4]-) and 1-Butyl-3-methylimidazolium tetrafluoroborate ([C4mim]+[BF4]-), into the one-step antisolvent Cs_{0.05}(FA_{0.83}MA_{0.17})_{0.95}Pb(I_{0.82}Br_{0.18})₃ perovskite films (the standard triple-cation double-halide perovskite composition).

These LAS additive perovskites were used to fabricate both single-junction and perovskite-silicon tandem solar cells (using TOPCon silicon bottom cells). The surfactant-containing precursor solutions were prepared by dissolving the same Cs_{0.05}(FA_{0.83}MA_{0.17})_{0.95}Pb(I_{0.82}Br_{0.18})₃ perovskite components in the desired molar ratios surfactant-containing DMF/DMSO mixed solvent, and all solutions were filtered (0.45 μm, PTFE) before use.

The authors perform a wide array of characterization techniques on the LAS additive perovskite films including PFQNM-AFM YM mapping, Raman (which is quite difficult to achieve in the perovskite field), ToF-SIMS, FTIR, DLS, XPS, GIWAX, and NMR. From the PFQNM-AFM YM mapping, which is the key characterization technique in this work, they were able to show that the residual tensile stress in the perovskite films can be significantly suppressed by the addition of the long-alkyl-chain [C4mim]+[C8SO4]-. The authors attribute this to the interaction of the LACs with the perovskite grains to "form a glue-like scaffold that effectively eliminates stresses by reducing the Young's modulus and thermal expansion coefficient". The results were also confirmed from GIXRD and Raman spectroscopy. The use of the PFQNM-AFM YM mapping to assess stress could be quite useful to the field of perovskites and result in more of its use if this work were published.

In terms of the segregation and micellization of the LASs, the authors first used ToF-SIMS to analyse their films with the LASs. The larger longer chain [C4mim]+[C8SO4]- additive sample saw an increase by one order of magnitude at the surface, indicating the accumulation of [C8SO4]- ions at the top surface of perovskite and the downward gradient distribution in the bulk film. This would be expected given the larger length chain of the [C4mim]+[C8SO4]- additive sample, although the authors may wish to refer to work from Harvey, S. P., et al. Investigating the Effects of Chemical Gradients on Performance and Reliability within Perovskite Solar Cells with TOF-SIMS. *Adv. Energy Mater.* 2020, 10, 1903674.

<https://doi.org/10.1002/aenm.201903674>, to examine common artifacts associated with the ToF-SIMS of perovskite films. The work then leads to the role of the additive segregation in the perovskite nucleation stage where the authors provide figures in the Supplementary Information to demonstrate this. They further study the LAS-modified perovskite films using the common UV-Vis absorption and PL (steady-state and time-resolved) techniques showing improvements in lifetime with the target LASs samples. What is interesting is they perform conductive AFM measurements to examine the difference between the grain interfaces and grain boundaries induced by the LASs, which showed that the current flowing through the perovskite grains is reduced from ~22 pA for the control to ~10 pA for the target samples.

They also run the stability testing of their unencapsulated single-junction and perovskite-silicon tandem solar cell devices up to the ISOS-2 standard (that is 65 °C, 50% RH, at AM1.5 sun illumination) – these stability measurement standardizations were outlined in the work by Khenkin, M.V. et al. “Consensus statement for stability assessment and reporting for perovskite photovoltaics based on ISOS procedures. *Nat. Energy* 5, 35-49 (2020)”, which has been referenced by the authors. They also provide the tracking of the temperature and relative humidity during the process that was provided in the Supplementary Information. The authors also had one of the devices certified by an independent third party, in this case, the Shanghai Institute of Microsystem and Information Technology (SIMIT). They claim that this is one the best stabilities achieved in an unencapsulated tandem device at MPPT under similar test conditions. They provide a Table (Supplementary Table 2) of existing literature for stability testing, which includes over 30 cases where stability of perovskite solar cells have been reported in literature.

In conclusion, the authors produce a very thorough scientific study with no major scientific concerns. The work is also novel, the LAS additives used would be received well by the perovskite field and thus the manuscript would be highly suitable for publication in *Nature Communications* with optional revisions. The quality of the scientific work that the authors have produced is of high-quality including the vast suite of characterization techniques (including in-depth explanations) that were used to analyse the perovskite films and devices with the LAS additives.

Major Concerns: None

Minor Concerns:

Introduction: The authors may wish to cite in their introduction a review on the topic of stress/strain, by Liu et al. Strain analysis and engineering in halide perovskite photovoltaics, *Nat Mater* 2021, 20(10) : 1337-1346. Or perhaps at line 314-315 may be more suitable here.

Line 146, how were the 3D elemental maps constructed? Was this from the ToF-SIMS measurement? It does not seem clear in the text and caption of Supplementary Fig. 8.

Line 212-215, the authors may wish to refer to seminal work by Jones et al. Energy Environ. Sci., 2019, 12, 596-606, here, which points to lattice strain directly associated with enhanced defect concentrations and non-radiative recombination, with strain patterns having a complex heterogeneity across multiple length scales.

Line 290 and Supplementary Figure 26, the term steady-state photocurrent output (SPO) seems a little confusing here (or likely a term that is less often used)? Is this the percentage conversion efficiency (PCE) or current density at MPPT of your measured devices?

Abbreviation is needed for DLS in the main text and Supplementary Information.

Grammatical Corrections:

Line 399 ...improve the perovskite crystallization but, more importantly, can also eliminate...

Line 407 ... and stable perovskite-based devices in the near future...

SI Line 28 ... perovskite film mainly originates from the differences in ...

Reviewer #2 (Remarks to the Author):

In this manuscript, Wang et al. systematically investigated a series of surfactants consisting of [C4mim]⁺ cation and sulfate anions with different alkyl chain lengths [C_nSO₄]⁻ on the material properties of perovskites and the device performance of ensuring single-junction and perovskite/silicon solar cells. The authors revealed beneficial effects of these long-chain anionic surfactants (LAS) in improving the perovskite crystallinity by modifying the nucleation and growth mechanism of the perovskite films. Moreover, they clearly demonstrate the role of [C4mim]⁺ [C8SO₄]⁻ in eliminating residual stresses in perovskite films by decreasing the Young's Modulus and thermal expansion coefficient. With these combined effects, the [C4mim]⁺ [C8SO₄]⁻ modified perovskite layers showed reduced film defects, suppressed ion migration, and improved interfacial energy level alignment, enabling the achievement of

high-efficiency single-junction and perovskite/silicon tandem solar cells with a PCE of 21.6% and 27%, respectively. More impressively, the [C4mim]⁺ [C8SO4]⁻ modified devices exhibited outstanding long-term operational stability, with the unencapsulated single-junction cell retaining over 85% of the original PCE after an MMT stressing test for 3000 h. Although there are some reports using ionic additives in perovskite solar cells, this comprehensive study on LAS-induced growth kinetics control of perovskite crystals, the mechanical properties, film stresses, and their effects on the devices, would be helpful for more advanced designs of efficient and stable perovskite-based single-junction and tandem solar cells. Overall, this is a high-quality manuscript that deserves high visibility and I would recommend it for publication in Nature Communications after addressing the following problems:

1. It would be better to also provide the corresponding AFM topography mapping of perovskite films in Figure 1a-d.
2. As the authors claimed that C0SO4 anions were gradient distributed in the bulk perovskite, mainly enriched at the buried interface. With this fact, the NiOX layer should be labeled in the middle rather than at the bottom of the 3D images (Figure S8a)? Please carefully check the label of the location for NiOX in the figure.
3. The authors provide some experimental evidence on the suppressed ion migrations with the LSAs. But, for the electrical poling experiments, there would be some confusing information originating from the samples or the fabricating procedures. The authors may need to provide some additional characterization results on the ion migration of the samples.
4. In the first paragraph of 'Reduction of residual stresses via LASs', the author chose the triple cation perovskite with a Br ratio of around 15%, which seems not consistent with the material used in the devices. Is there any typo in labeling the material composition?
5. As the LASs consisting of [CnSO4]⁻ featuring different alkyl chain lengths induced obvious differences in the perovskite films, I am curious how the solar cells by incorporating different [CnSO4]⁻ would behave?
6. The authors claimed 'reduction of nonradiative recombination' based on the increased PL lifetime, which is not solid enough. It is suggested to also provide the PLQY measurement results of the films to support the claim.
7. The authors provide a detailed explanation of the upshifted VBM, while the upshifted Fermi level was somehow ignored. Considering the important role of the surface properties of perovskite film in p-i-n solar cells, the LAS-induced n-type doping should be further explained.
8. The authors provided detailed evolution of the performance parameters during the ISOS-L2 measurement, I am wondering if the device hysteresis has also changed with the device degradation. That would be more clear if the authors could provide some scanned J-V curves and the SPO results at different intervals during the measurement.
9. There are some additional typos that need to be carefully checked and revised, e.g. "residue stresses", "Young's modulus"

Reviewer #3 (Remarks to the Author):

The authors have introduced a series of long-alkyl-chain anionic surfactant additives to the perovskite precursor, and studies the effect of the alkyl chain length on the perovskite crystallization kinetics and thus the perovskite film optoelectronic and mechanic properties, as well as the as-prepared perovskite and tandem solar cell performance, including both efficiency and stability. The long alkyl chain anionic surfactant is based on [C4mim]+[CnSO4]-, for which the cation has been previously introduced to perovskite solar cell (PSC) but the introduction of the anion part is for the first time being reported, highlighting the novelty of the work. Longer alkyl chain (n=8) was found to be beneficial to reduce the perovskite film residual stress and thus improve the as-prepared device efficiency and stability.

This work has performed comprehensive characterisation at both material and device levels and provided sounded and robust data interpretation. However, an important question to be pointed out is the inconsistency between the perovskite film characterisation and the device testing. For perovskite film characterisation, various surfactants with different alkyl chain length (n=1, 4, 8) as well as different anion (BF4-) were compared to assess their impact of perovskite film quality. However, only one surfactant [C4mim]+[C8SO4]- was tested and reported at the device level. This makes it incomplete to understand the correlation between the film optoelectronic property and the device performance. Hence, it is strongly suggested that all the surfactant molecules are tested at device level. This is particularly important given the fact that the device performance (21%) reported in this work still lags behind that of the state-of-the-art (>24%) with the same bandgap. This raises the question whether such surfactant could also benefit the state-of-the-art perovskite film and device.

Another technical question to answer is that the authors also observed increased YMs for short-chain [C4mim]+[BF4] –additive. This seems to be contradictory to the report by Bai, S. et al [Ref. 30, Planar perovskite solar cells with long-term stability using ionic liquid additives. Nature 571, 245-250 (2019).], where C4mim]+[BF4] – was found to improve the PSC efficiency and stability. Therefore, in this work, the authors are suggested to also test the device performance for the PSC treated with [C4mim]+[BF4] – for comparison.

The fabrication process of how to introduce the surfactant into the perovskite film is not clear. In the experimental part, for the single-junction PSC, the description of “...the surfactant-containing precursor solutions were prepared by dissolving the same Cs0.05(FA0.83MA0.17)0.95Pb(I0.82Br0.18)3 perovskite components in the desired molar ratios surfactant-containing DMF/DMSO mixed solvent, and all solutions were filtered (0.45 μm, 431 PTFE) before use” does not read clear. For the tandem solar cell, no description about how to incorporate the surfactant can be found.

The light stability of the tandem solar cell was reported to be much worse than that of the single-junction PSC. The authors pointed out one potential reason which is “the free carries accumulated at the

poor ETL/perovskite interface will reduce the ion migration activation energy and then accelerate the perovskite degradation,” Is there any evidence for this hypothesis? In addition, there is prominent current mismatch between the Si sub-cell and the perovskite sub-cell according to EQE of the tandem device. Can the authors comment on the impact of the current mismatch on the stability of the tandem device?

REVIEWER COMMENTS

Reviewer #1 (Remarks to the Author):

The manuscript entitled "Segregation and micellization of long-chain anionic surfactants enabling stable perovskite/silicon tandems with greatly suppressed stress corrosion" with Ref #: "NCOMMS-22-46934", by authors "Xinlong Wang, Zhiqin Ying, Jichun Ye et al." is a research article on the employment of long-alkyl-chain anionic surfactant (LAS) additives: 1-Butyl-3-methylimidazolium hydrogen sulfate ($[\text{C}_4\text{mim}]^+[\text{C}_0\text{SO}_4]^-$), 1-Butyl-3-methylimidazolium methyl sulfate ($[\text{C}_4\text{mim}]^+[\text{C}_1\text{SO}_4]^-$), 1-Butyl-3-methylimidazolium octyl sulfate ($[\text{C}_4\text{mim}]^+[\text{C}_8\text{SO}_4]^-$) and 1-Butyl-3-methylimidazolium tetrafluoroborate ($[\text{C}_4\text{mim}]^+[\text{BF}_4]^-$), into the one-step antisolvent $\text{Cs}_{0.05}(\text{FA}_{0.83}\text{MA}_{0.17})_{0.95}\text{Pb}(\text{I}_{0.82}\text{Br}_{0.18})_3$ perovskite films (the standard triple-cation double-halide perovskite composition).

These LAS additive perovskites were used to fabricate both single-junction and perovskite-silicon tandem solar cells (using TOPCon silicon bottom cells). The surfactant-containing precursor solutions were prepared by dissolving the same $\text{Cs}_{0.05}(\text{FA}_{0.83}\text{MA}_{0.17})_{0.95}\text{Pb}(\text{I}_{0.82}\text{Br}_{0.18})_3$ perovskite components in the desired molar ratios surfactant-containing DMF/DMSO mixed solvent, and all solutions were filtered (0.45 μm , PTFE) before use.

The authors perform a wide array of characterization techniques on the LAS additive perovskite films including PFQNM-AFM YM mapping, Raman (which is quite difficult to achieve in the perovskite field), ToF-SIMS, FTIR, DLS, XPS, GIWAX, and NMR. From the PFQNM-AFM YM mapping, which is the key characterization technique in this work, they were able to show that the residual tensile stress in the perovskite films can be significantly suppressed by the addition of the long-alkyl-chain $[\text{C}_4\text{mim}]^+[\text{C}_8\text{SO}_4]^-$. The authors attribute this to the interaction of the LACs with the perovskite grains to "form a glue-like scaffold that effectively eliminates stresses by reducing the Young's modulus and thermal expansion coefficient". The results were also confirmed from GIXRD and Raman spectroscopy. **The use of the PFQNM-AFM**

YM mapping to assess stress could be quite useful to the field of perovskites and result in more of its use if this work were published.

In terms of the segregation and micellization of the LASs, the authors first used ToF-SIMS to analyse their films with the LASs. The larger longer chain $[C_4mim]^+[C_8SO_4]^-$ additive sample saw an increase by one order of magnitude at the surface, indicating the accumulation of $[C_8SO_4]^-$ ions at the top surface of perovskite and the downward gradient distribution in the bulk film. This would be expected given the larger length chain of the $[C_4mim]^+[C_8SO_4]^-$ additive sample, although the authors may wish to refer to work from Harvey, S. P., et al. Investigating the Effects of Chemical Gradients on Performance and Reliability within Perovskite Solar Cells with TOF-SIMS. *Adv. Energy Mater.* 2020, 10, 1903674. <https://doi.org/10.1002/aenm.201903674>, to examine common artifacts associated with the ToF-SIMS of perovskite films. The work then leads to the role of the additive segregation in the perovskite nucleation stage where the authors provide figures in the Supplementary Information to demonstrate this. They further study the LAS-modified perovskite films using the common UV-Vis absorption and PL (steady-state and time-resolved) techniques showing improvements in lifetime with the target LASs samples. What is interesting is they perform conductive AFM measurements to examine the difference between the grain interfaces and grain boundaries induced by the LASs, which showed that the current flowing through the perovskite grains is reduced from ~22 pA for the control to ~10 pA for the target samples.

They also run the stability testing of their unencapsulated single-junction and perovskite-silicon tandem solar cell devices up to the ISOS-2 standard (that is 65 °C, 50% RH, at AM1.5 sun illumination) – these stability measurement standardizations were outlined in the work by Khenkin, M.V. et al. “Consensus statement for stability assessment and reporting for perovskite photovoltaics based on ISOS procedures. *Nat. Energy* 5, 35-49 (2020)”, which has been referenced by the authors. They also provide the tracking of the temperature and relative humidity during the process that was provided in the Supplementary Information. The authors also had one of the devices certified by an independent third party, in this case, the Shanghai Institute of

Microsystem and Information Technology (SIMIT). They claim that this is one the best stabilities achieved in an unencapsulated tandem device at MPPT under similar test conditions. They provide a Table (Supplementary Table 2) of existing literature for stability testing, which includes over 30 cases where stability of perovskite solar cells have been reported in literature.

In conclusion, the authors produce a very thorough scientific study with no major scientific concerns. The work is also novel, the LAS additives used would be received well by the perovskite field and thus the manuscript would be highly suitable for publication in Nature Communications with optional revisions. The quality of the scientific work that the authors have produced is of high-quality including the vast suite of characterization techniques (including in-depth explanations) that were used to analyse the perovskite films and devices with the LAS additives.

Major Concerns: None

Minor Concerns:

- 1. Comment:** Introduction: The authors may wish to cite in their introduction a review on the topic of stress/strain, by Liu et al. Strain analysis and engineering in halide perovskite photovoltaics, Nat Mater 2021, 20(10): 1337-1346. Or perhaps at line 314-315 may be more suitable here.
- 2. Comment:** Line 146, how were the 3D elemental maps constructed? Was this from the ToF-SIMS measurement? It does not seem clear in the text and caption of Supplementary Fig. 8.
- 3. Comment:** Line 212-215, the authors may wish to refer to seminal work by Jones et al. Energy Environ. Sci., 2019, 12, 596-606, here, which points to lattice strain directly associated with enhanced defect concentrations and non-radiative recombination, with strain patterns having a complex heterogeneity across multiple length scales.

4. Comment: Line 290 and Supplementary Figure 26, the term steady-state photocurrent output (SPO) seems a little confusing here (or likely a term that is less often used)? Is this the percentage conversion efficiency (PCE) or current density at MPPT of your measured devices?

5. Comment: Abbreviation is needed for DLS in the main text and Supplementary Information.

6. Comment: Grammatical Corrections:

Line 399 ...improve the perovskite crystallization but, more importantly, can also eliminate...

Line 407 ... and stable perovskite-based devices in the near future...

SI Line 28 ... perovskite film mainly originates from the differences in ...

Reviewer #2 (Remarks to the Author):

In this manuscript, Wang et al. systematically investigated a series of surfactants consisting of $[\text{C}_4\text{mim}]^+$ cation and sulfate anions with different alkyl chain lengths $[\text{C}_n\text{SO}_4]^-$ on the material properties of perovskites and the device performance of ensuring single-junction and perovskite/silicon solar cells. The authors revealed beneficial effects of these long-chain anionic surfactants (LAS) in improving the perovskite crystallinity by modifying the nucleation and growth mechanism of the perovskite films. Moreover, they clearly demonstrate the role of $[\text{C}_4\text{mim}]^+[\text{C}_8\text{SO}_4]^-$ in eliminating residual stresses in perovskite films by decreasing the Young's Modulus and thermal expansion coefficient. With these combined effects, the $[\text{C}_4\text{mim}]^+[\text{C}_8\text{SO}_4]^-$ modified perovskite layers showed reduced film defects, suppressed ion migration, and improved interfacial energy level alignment, enabling the achievement of high-efficiency single-junction and perovskite/silicon tandem solar cells with a PCE of 21.6% and 27%, respectively. More impressively, the $[\text{C}_4\text{mim}]^+[\text{C}_8\text{SO}_4]^-$ modified devices exhibited outstanding long-term operational stability, with the unencapsulated single-

junction cell retaining over 85% of the original PCE after an MMP stressing test for 3000 h. Although there are some reports using ionic additives in perovskite solar cells, this comprehensive study on LAS-induced growth kinetics control of perovskite crystals, the mechanical properties, film stresses, and their effects on the devices, would be helpful for more advanced designs of efficient and stable perovskite-based single-junction and tandem solar cells. **Overall, this is a high-quality manuscript that deserves high visibility and I would recommend it for publication in Nature Communications after addressing the following problems:**

- 1. Comment:** It would be better to also provide the corresponding AFM topography mapping of perovskite films in Figure 1a-d.
- 2. Comment:** As the authors claimed that C_0SO_4 anions were gradient distributed in the bulk perovskite, mainly enriched at the buried interface. With this fact, the NiO_x layer should be labeled in the middle rather than at the bottom of the 3D images (Figure S8a)? Please carefully check the label of the location for NiO_x in the figure.
- 3. Comment:** The authors provide some experimental evidence on the suppressed ion migrations with the LSAs. But, for the electrical poling experiments, there would be some confusing information originating from the samples or the fabricating procedures. The authors may need to provide some additional characterization results on the ion migration of the samples.
- 4. Comment:** In the first paragraph of ‘Reduction of residual stresses via LASs’, the author chose the triple cation perovskite with a Br ratio of around 15%, which seems not consistent with the material used in the devices. Is there any typo in labeling the material composition?

5. **Comment:** As the LASs consisting of $[C_nSO_4]^-$ featuring different alkyl chain lengths induced obvious differences in the perovskite films, I am curious how the solar cells by incorporating different $[C_nSO_4]^-$ would behave?
6. **Comment:** The authors claimed ‘reduction of nonradiative recombination’ based on the increased PL lifetime, which is not solid enough. It is suggested to also provide the PLQY measurement results of the films to support the claim.
7. **Comment:** The authors provide a detailed explanation of the upshifted VBM, while the upshifted Fermi level was somehow ignored. Considering the important role of the surface properties of perovskite film in p-i-n solar cells, the LAS-induced n-type doping should be further explained.
8. **Comment:** The authors provided detailed evolution of the performance parameters during the ISOS-L2 measurement, I am wondering if the device hysteresis has also changed with the device degradation. That would be more clear if the authors could provide some scanned J–V curves and the SPO results at different intervals during the measurement.
9. **Comment:** There are some additional typos that need to be carefully checked and revised, e.g. “residue stresses”, “Young’s modulus”

Reviewer #3 (Remarks to the Author):

The authors have introduced a series of long-alkyl-chain anionic surfactant additives to the perovskite precursor, and studies the effect of the alkyl chain length on the perovskite crystallization kinetics and thus the perovskite film optoelectronic and mechanic properties, as well as the as-prepared perovskite and tandem solar cell performance, including both efficiency and stability. The long alkyl chain anionic surfactant is based on $[C_4mim]^+[C_nSO_4]^-$, for which the cation has been previously

introduced to perovskite solar cell (PSC) **but the introduction of the anion part is for the first time being reported, highlighting the novelty of the work.** Longer alkyl chain (n=8) was found to be beneficial to reduce the perovskite film residual stress and thus improve the as-prepared device efficiency and stability.

1. Comment: This work has performed comprehensive characterisation at both material and device levels and provided sounded and robust data interpretation. However, an important question to be pointed out is the inconsistency between the perovskite film characterisation and the device testing. For perovskite film characterisation, various surfactants with different alkyl chain length (n=0, 1, 8) as well as different anion (BF_4^-) were compared to assess their impact of perovskite film quality. However, only one surfactant $[\text{C}_4\text{mim}]^+[\text{C}_8\text{SO}_4]^-$ was tested and reported at the device level. This makes it incomplete to understand the correlation between the film optoelectronic property and the device performance. Hence, it is strongly suggested that all the surfactant molecules are tested at device level. This is particularly important given the fact that the device performance (21%) reported in this work still lags behind that of the state-of-the-art (>24%) with the same bandgap. This raises the question whether such surfactant could also benefit the state-of-the-art perovskite film and device.

2. Comment: Another technical question to answer is that the authors also observed increased YMs for short-chain $[\text{C}_4\text{mim}]^+[\text{BF}_4]^-$ additive. This seems to be contradictory to the report by Bai, S. et al [Ref. 30, Planar perovskite solar cells with long-term stability using ionic liquid additives. Nature 571, 245-250 (2019).], where $[\text{C}_4\text{mim}]^+[\text{BF}_4]^-$ was found to improve the PSC efficiency and stability. Therefore, in this work, the authors are suggested to also test the device performance for the PSC treated with $[\text{C}_4\text{mim}]^+[\text{BF}_4]^-$ for comparison.

3. Comment: The fabrication process of how to introduce the surfactant into the perovskite film is not clear. In the experimental part, for the single-junction PSC, the description of "...the surfactant-containing precursor solutions were prepared by dissolving the same $\text{Cs}_{0.05}(\text{FA}_{0.83}\text{MA}_{0.17})_{0.95}\text{Pb}(\text{I}_{0.82}\text{Br}_{0.18})_3$ perovskite components in the desired molar ratios surfactant-containing DMF/DMSO mixed solvent, and all solutions were filtered (0.45 μm , 431 PTFE) before use" does not read clear. For the tandem solar cell, no description about how to incorporate the surfactant can be found.

4. Comment: The light stability of the tandem solar cell was reported to be much worse than that of the single-junction PSC. The authors pointed out one potential reason which is "the free carriers accumulated at the poor ETL/perovskite interface will reduce the ion migration activation energy and then accelerate the perovskite degradation," Is there any evidence for this hypothesis? In addition, there is prominent current mismatch between the Si sub-cell and the perovskite sub-cell according to the EQE of the tandem device. Can the authors comment on the impact of the current mismatch on the stability of the tandem device?

List of point-to-point response of reviews' comments

Reviewer #1 (Remarks to the Author):

The manuscript entitled "Segregation and micellization of long-chain anionic surfactants enabling stable perovskite/silicon tandems with greatly suppressed stress corrosion" with Ref #: "NCOMMS-22-46934", by authors "Xinlong Wang, Zhiqin Ying, Jichun Ye et al." is a research article on the employment of long-alkyl-chain anionic surfactant (LAS) additives: 1-Butyl-3-methylimidazolium hydrogen sulfate ($[\text{C}_4\text{mim}]^+[\text{C}_0\text{SO}_4]^-$), 1-Butyl-3-methylimidazolium methyl sulfate ($[\text{C}_4\text{mim}]^+[\text{C}_1\text{SO}_4]^-$), 1-Butyl-3-methylimidazolium octyl sulfate ($[\text{C}_4\text{mim}]^+[\text{C}_8\text{SO}_4]^-$) and 1-Butyl-3-methylimidazolium tetrafluoroborate ($[\text{C}_4\text{mim}]^+[\text{BF}_4]^-$), into the one-step antisolvent $\text{Cs}_{0.05}(\text{FA}_{0.83}\text{MA}_{0.17})_{0.95}\text{Pb}(\text{I}_{0.82}\text{Br}_{0.18})_3$ perovskite films (the standard triple-cation double-perovskite composition).

These LAS additive perovskites were used to fabricate both single-junction and perovskite-silicon tandem solar cells (using TOPCon silicon bottom cells). The surfactant-containing precursor solutions were prepared by dissolving the same $\text{Cs}_{0.05}(\text{FA}_{0.83}\text{MA}_{0.17})_{0.95}\text{Pb}(\text{I}_{0.82}\text{Br}_{0.18})_3$ perovskite components in the desired molar ratios surfactant-containing DMF/DMSO mixed solvent, and all solutions were filtered (0.45 μm , PTFE) before use.

The authors perform a wide array of characterization techniques on the LAS additive perovskite films including PFQNM-AFM YM mapping, Raman (which is quite difficult to achieve in the perovskite field), ToF-SIMS, FTIR, DLS, XPS, GIWAX, and NMR. From the PFQNM-AFM YM mapping, which is the key characterization technique in this work, they were able to show that the residual tensile stress in the perovskite films can be significantly suppressed by the addition of the long-alkyl-chain $[\text{C}_4\text{mim}]^+[\text{C}_8\text{SO}_4]^-$. The authors attribute this to the interaction of the LACs with the perovskite grains to "form a glue-like scaffold that effectively eliminates stressed by reducing the Young's modulus and thermal expansion coefficient". The results were also confirmed from GIXRD and Raman spectroscopy. **The use of the PFQNM-AFM**

YM mapping to assess stress could be quite useful to the field of perovskites and result in more of its use if this work were published.

In terms of the segregation and micellization of the LASs, the authors first used ToF-SIMS to analyse their films with the LASs. The larger longer chain $[\text{C}_4\text{mim}]^+[\text{C}_8\text{SO}_4]^-$ additive sample saw an increase by one order of magnitude at the surface, indicating the accumulation of $[\text{C}_8\text{SO}_4]^-$ ions at the top surface of perovskite and the downward gradient distribution in the bulk film. This would be expected given the larger length chain of the $[\text{C}_4\text{mim}]^+[\text{C}_8\text{SO}_4]^-$ additive sample, although the authors may wish to refer to work from Harvey, S. P., et al. Investigating the Effects of Chemical Gradients on Performance and Reliability within Perovskite Solar Cells with TOF-SIMS. *Adv. Energy Mater.* 2020, 10, 1903674. <https://doi.org/10.1002/aenm.201903674>, to examine common artifacts associated with the ToF-SIMS of perovskite films. The work then leads to the role of the additive segregation in the perovskite nucleation stage where the authors provide figures in the Supplementary Information to demonstrate this. They further study the LAS-modified perovskite films using the common UV-Vis absorption and PL (steady-state and time-resolved) techniques showing improvements in lifetime with the target LASs samples. What is interesting is they perform conductive AFM measurements to examine the difference between the grain interfaces and grain boundaries induced by the LASs, which showed that the current flowing through the perovskite grains is reduced from ~22 pA for the control to ~10 pA for the target samples.

They also run the stability testing of their unencapsulated single-junction and perovskite-silicon tandem solar cell devices up to the ISOS-2 standard (that is 65 °C, 50% RH, at AM1.5 sun illumination) – these stability measurement standardizations were outlined in the work by Khenkin, M.V. et al. “Consensus statement for stability assessment and reporting for perovskite photovoltaics based on ISOS procedures. *Nat. Energy* 5, 35-49 (2020)”, which has been referenced by the authors. They also provide the tracking of the temperature and relative humidity during the process that was provided in the Supplementary Information. The authors also had one of the devices certified by an independent third party, in this case, the Shanghai Institute of

Microsystem and Information Technology (SIMIT). They claim that this is one the best stabilities achieved in an unencapsulated tandem device at MPPT under similar test conditions. They provide a Table (Supplementary Table 2) of existing literature for stability testing, which includes over 30 cases where stability of perovskite solar cells have been reported in literature.

In conclusion, the authors produce a very thorough scientific study with no major scientific concerns. The work is also novel, the LAS additives used would be received well by the perovskite field and thus the manuscript would be highly suitable for publication in Nature Communications with optional revisions. The quality of the scientific work that the authors have produced is of high-quality including the vast suite of characterization techniques (including in-depth explanations) that were used to analyse the perovskite films and devices with the LAS additives.

Major Concerns: None

Minor Concerns:

Our Response:

Thank you for your high appraisal of this work. We are of great gratitude for your instructive comments and positive appraisal.

- 1. Comment:** Introduction: The authors may wish to cite in their introduction a review on the topic of stress/strain, by Liu et al. Strain analysis and engineering in halide perovskite photovoltaics, Nat Mater 2021, 20(10): 1337-1346. Or perhaps at line 314-315 may be more suitable here.

Our Reply:

Thank you for your good suggestion.

Our Response:

On page 9, line 336-338, the reference had been cited as references [9] in the revised manuscript: "However, it is well known that both defect passivation and stress reduction can synergistically contribute to the perovskite stability^{9,64}."

2. Comment: Line 146, how were the 3D elemental maps constructed? Was this from the ToF-SIMS measurement? It does not seem clear in the text and caption of Supplementary Fig. 8.

Our Reply:

Thanks for pointing out this unclear information. The 3D elemental maps were directly constructed from the ToF-SIMS measurements. It is well known that the ToF-SIMS enables 1D depth profiling, 2D lateral imaging, and 3D tomography, with a depth resolution of less than 1 nm and a lateral spatial resolution of less than 100 nm [*Adv Funct Mater* 30, 2002201 (2020). 10.1002/adfm.202002201]. Dynamic SIMS with high primary ion doses and rapid erosion rates can yield chemical information both laterally and vertically, thus allowing 3D analysis. Here, when the depth profiling is combined with high-resolution imaging, the 3D tomography can be realized (each 3D reconstruction is $50 \times 50 \times 0.5 \mu\text{m}$), which can yield insight into changes of cation/anion gradients and lateral distribution upon changes in processing [*Adv Energy Mater* 10, 1903674 (2020). 10.1002/aenm.201903674].

Our Response:

The related discussion: "The 3D ToF-SIMS tomography is realized by combining the depth profiling with high-resolution imaging (each 3D reconstruction is $50 \times 50 \times 0.5 \mu\text{m}$)." had been added in the caption of Supplementary Fig. 9.

3. Comment: Line 212-215, the authors may wish to refer to seminal work by Jones et al. *Energy Environ. Sci.*, 2019, 12, 596-606, here, which points to lattice strain directly associated with enhanced defect concentrations and non-radiative recombination, with strain patterns having a complex heterogeneity across multiple length scales.

Our Reply:

Thank you for your introduction to this seminal work. In this reference, the authors showed that defects are related to complicated strain patterns that appear on multiple length scales in perovskite films-ranging from tens of micrometers down

to the tens of nanometer scale. This work has profound implications for our understanding of the operation of the materials on the micro-scale [*Energy Environ Sci* 12, 596-606 (2019).10.1039/C8EE02751J].

Our Response:

On page 6, line 214-216, the work had been cited as references [47] in the revised manuscript: "At first sight, the enlarged grain sizes here can readily improve the mechanical stability of the perovskite films by reducing the defect-laden GBs, which are the weakest mechanical links in polycrystalline films⁴⁷."

4. Comment: Page 290 and Supplementary Figure 26, the term steady-state photocurrent output (SPO) seems a little confusing here (or likely a term that is less often used)? Is this the percentage conversion efficiency (PCE) or current density at MPPT of your measured devices?

Our Reply:

We are sorry for this confusing expression. The term steady-state photocurrent output (SPO) used in the manuscript means the stabilized current density at a bias voltage near the maximum power point (*Energy Environ Sci* 15, 244-253 (2022).10.1039/D1EE01778K). To more clearly express the meaning, we revised the "steady-state photocurrent output (SPO)" into the commonly used "stabilized power output (SPO)". [*Nature* 611, 278-283 (2022).10.1038/s41586-022-05268-x; *Science* 377, 302-306 (2022). 10.1126/science.abn8910]

Our Response:

On page 8, line 307-309, we have revised the sentence from "... steady-state photocurrent output (SPO) curves of the best-performing ..." to "... stabilized power output (SPO) curves of the best-performing ..."

5. Comment: Abbreviation is needed for DLS in the main text and Supplementary Information.

Our Reply:

Thanks for your keen observation and kind reminders! We had checked the

WHOLE manuscript carefully and tried to avoid the error of abbreviating without the full name comment.

Our Response:

In the section of **Methods**, "DLS measurements were performed by a Zetasizer Nano ZS instrument with a 633-nm He-Ne laser." had been revised to "**Dynamic light scattering (DLS) measurements were performed by a Zetasizer Nano ZS instrument with a 633-nm He-Ne laser.**"

In **Supplementary Fig. 14**, "DLS measurements of control and $[\text{C}_4\text{mim}]^+[\text{C}_8\text{SO}_4]^-$ treated perovskite precursors." had been revised to "**Dynamic light scattering (DLS) measurements of control and $[\text{C}_4\text{mim}]^+[\text{C}_8\text{SO}_4]^-$ treated perovskite precursors.**"

6. Comment: Grammatical Corrections:

Line 399 ...improve the perovskite crystallization but, more importantly, can also eliminate...

Line 407 ... and stable perovskite-based devices in the near future...

SI Line 28 ... perovskite film mainly originates from the differences in ...

Our Reply:

Thanks a lot for pointing out these mistakes.

Our Response:

On **page 11, line 435-437**, "...more importantly, **also can** eliminate the film residual stresses." had been revised to "...more importantly, **can also** eliminate the film residual stresses."

On **page 12, line 443-444**, "...towards the commercial production of efficient and stable perovskite-based devices **in near future**" had been revised to "...towards the commercial production of efficient and stable perovskite-based devices **in the near future.**".

On **SI page 2, line 28-30**, "The stress in the perovskite film mainly **origins from** the differences ..." had been revised to "The stress in the perovskite film mainly **originates from** the differences ...".

Reviewer #2 (Remarks to the Author):

In this manuscript, Wang et al. systematically investigated a series of surfactants consisting of $[C_4mim]^+$ cation and sulfate anions with different alkyl chain lengths $[C_nSO_4]^-$ on the material properties of perovskites and the device performance of ensuring single-junction and perovskite/silicon solar cells. The authors revealed beneficial effects of these long-chain anionic surfactants (LAS) in improving the perovskite crystallinity by modifying the nucleation and growth mechanism of the perovskite films. Moreover, they clearly demonstrate the role of $[C_4mim]^+[C_8SO_4]^-$ in eliminating residual stresses in perovskite films by decreasing the Young's Modulus and thermal expansion coefficient. With these combined effects, the $[C_4mim]^+[C_8SO_4]^-$ modified perovskite layers showed reduced film defects, suppressed ion migration, and improved interfacial energy level alignment, enabling the achievement of high-efficiency single-junction and perovskite/silicon tandem solar cells with a PCE of 21.6% and 27%, respectively. More impressively, the $[C_4mim]^+[C_8SO_4]^-$ modified devices exhibited outstanding long-term operational stability, with the unencapsulated single-junction cell retaining over 85% of the original PCE after an MMP stressing test for 3000 h. Although there are some reports using ionic additives in perovskite solar cells, this comprehensive study on LAS-induced growth kinetics control of perovskite crystals, the mechanical properties, film stresses, and their effects on the devices, would be helpful for more advanced designs of efficient and stable perovskite-based single-junction and tandem solar cells. **Overall, this is a high-quality manuscript that deserves high visibility and I would recommend it for publication in Nature Communications after addressing the following problems:**

Our Response:

Thank you very much for your positive and insightful comments and your recognition of the significance of our work! The related comments will be addressed one by one below:

- 1. Comment:** It would be better to also provide the corresponding AFM topography mapping of perovskite films in Figure 1a-d.

Our Reply:

Thanks for your valuable suggestions. **Figure R1** shows the corresponding AFM topography mapping of perovskite films in Figure 1a-d. Compared with the control film, the perovskite films exhibit a slightly decreased grain size after the incorporation of $[\text{C}_4\text{mim}]^+[\text{C}_0\text{SO}_4]^-$ and $[\text{C}_4\text{mim}]^+[\text{C}_1\text{SO}_4]^-$. On the contrary, the grains became larger for the $[\text{C}_4\text{mim}]^+[\text{C}_8\text{SO}_4]^-$ -based perovskite film. The enlarged perovskite grain size can also be confirmed by the scanning electron microscopy (SEM) results (Supplementary Fig. 19).

Figure R1. AFM topography mapping of perovskite thin films without (referred as control sample) (a) additives, $[\text{C}_4\text{mim}]^+[\text{C}_0\text{SO}_4]^-$ (b), $[\text{C}_4\text{mim}]^+[\text{C}_1\text{SO}_4]^-$ (c) and $[\text{C}_4\text{mim}]^+[\text{C}_8\text{SO}_4]^-$ (d) treated films.

Our Response:

The corresponding AFM topography mapping of perovskite films in Figure 1a-d had been added in **Supplementary Fig. 2**.

2. **Comment:** As the authors claimed that C_0SO_4 anions were gradient distributed in the bulk perovskite, mainly enriched at the buried interface. With this fact, the NiO_x layer should be labeled in the middle rather than at the bottom of the 3D images (Figure S8a)? Please carefully check the label of the location for NiO_x in the figure.

Our Reply:

Thanks a lot for pointing out these mistakes. The location of the NiO_x layer after correction in the reconstructed 3D images had been shown in **Figure R2**.

Figure R2. The reconstructed 3D images of (a) $[C_0SO_4]^-$ and (b) $[C_8SO_4]^-$ in the $[C_4mim]^+[C_0SO_4]^-$ treated and $[C_4mim]^+[C_8SO_4]^-$ treated perovskite film, respectively.

Our Response:

In **Supplementary Figure 9**, the location of the NiO_x layer in the 3D images had been marked correctly.

3. **Comment:** The authors provide some experimental evidence on the suppressed ion migrations with the LSAs. But, for the electrical poling experiments, there would be some confusing information originating from the samples or the fabricating procedures. The authors may need to provide some additional characterization results on the ion migration of the samples.

Our Reply:

We clearly recognize the concern raised by the reviewer. We further tested the activation energy (Ea) of ion migration through the conductivity measurements of perovskite films under different temperatures [*Nat Photonics* 16, 588-594 (2022).10.1038/s41566-022-01033-8; *Science* 365, 473-478 (2019). 10.1126/science.aax3294]. Based on the Arrhenius plot shown in **Figure R3a**, we can extract an Ea of 0.142 eV for control films, which is comparable with the measured value in previously reported perovskites [*Nat Photonics* 16, 588-594 (2022).10.1038/s41566-022-01033-8]. By contrast, we observe a nearly double Ea (0.252 eV) in target films, proving that halide migration has been substantially suppressed in the target films. This result can also be confirmed by the transient ionic current (TIC) measurements (**Figure R3b**) as the reduced mobile ion concentrations in the target samples ($3.08 \times 10^{18} \text{ cm}^{-3}$) compared with control samples ($9.26 \times 10^{18} \text{ cm}^{-3}$). The concentrations of mobile ions in the perovskite films were estimated from the TIC based on the following formula [*Energy Environ Sci* 8, 1256-1260 (2015).10.1039/C4EE04064C]:

$$n = \frac{\int_{t_1}^{t_2} J dt}{eL}$$

where t is the time, J is the current density, e is the elementary charge and L is the thickness of the perovskite film. The transient ionic relaxation current was conducted at dark with an external forward bias at V_{OC} for 60 s. The measured current is mainly given by the redistribution of mobile ions in the perovskite layer after removing the external applied voltage because of the dark test condition. It is also worth noting that electrical poling experiments mentioned in this paper has been widely accepted as a means to characterize the ion migration of perovskite films [*Adv Energy Mater* 9, 1901852 (2019). 10.1002/aenm.201901852; *Nat Mater* 14, 193-198 (2015).10.1038/nmat4150; *Adv Mater* 30, 1707350 (2018). 10.1002/adma.201707350]. These results confirmed the suppressed ion migrations in the LSAs modified perovskite films, suggesting the similar results from the electrical poling experiments.

Figure R3. (a) Temperature-dependent conductivity of control and target perovskite films. (b) Transient ionic current (TIC) for extracting mobile ion concentration within the perovskite films of the control and target devices.

Our Response:

The measurements of temperature-dependent conductivity and transient ionic current had been added in Supplementary Figs. 23 and 24.

On page 7, line 263-271, the corresponding discussion: "To further verify this, a temperature-dependent conductivity measurement was performed to obtain the activation energy (E_a) for ion migration. Based on the Arrhenius plot shown in Supplementary Fig. 23, we can extract an E_a of 0.142 eV for control films, which is comparable with the value in previously reported perovskites. By contrast, we observe a nearly double E_a (0.252 eV) in target films, proving that halide migration has been substantially suppressed in the target films. This result can also be confirmed by the transient ionic current (TIC) measurements (Supplementary Fig. 24) as the reduced mobile ion concentrations in the target samples ($3.08 \times 10^{18} \text{ cm}^{-3}$) compared with control samples ($9.26 \times 10^{18} \text{ cm}^{-3}$)." had been added.

4. Comment: In the first paragraph of 'Reduction of residual stresses via LASs', the author chose the triple cation perovskite with a Br ratio of around 15%, which seems not consistent with the material used in the devices. Is there any typo in labeling the material composition?

Our Reply:

Thank the reviewer for pointing out the mistake.

Our Response:

On page 3, line 83-87, we have revised the sentence from "We chose the triple cation perovskite, $\text{Cs}_{0.05}(\text{FA}_{0.83}\text{MA}_{0.17})_{0.95}\text{Pb}(\text{I}_{0.85}\text{Br}_{0.15})_3$ with a bandgap of 1.63 eV ..." to "We chose the triple cation perovskite, $\text{Cs}_{0.05}(\text{FA}_{0.83}\text{MA}_{0.17})_{0.95}\text{Pb}(\text{I}_{0.82}\text{Br}_{0.18})_3$ with a bandgap of 1.63 eV ...".

5. Comment: As the LASs consisting of $[\text{C}_n\text{SO}_4]^-$ featuring different alkyl chain lengths induced obvious differences in the perovskite films, I am curious how the solar cells by incorporating different $[\text{C}_n\text{SO}_4]^-$ would behave?

Our Reply:

Thanks for your precious comments and suggestions. We first performed J–V and SPO measurements for the devices based on different $[\text{C}_n\text{SO}_4]^-$. As shown in Figure R4, the PCE improvements were mainly the result of enhanced V_{OC} and FF in LASs-treated devices. The improvement of $[\text{C}_4\text{mim}]^+[\text{C}_8\text{SO}_4]^-$ -treated devices was the most significant, while the $[\text{C}_4\text{mim}]^+[\text{C}_0\text{SO}_4]^-$ -treated devices were accompanied with an obvious hysteresis. These differences can be attributed to the different distribution of additives in perovskite films [Nano Energy 97, 107193 (2022). 10.1016/j.nanoen.2022.107193].

Figure R4. Performance parameters distribution for the control, $[\text{C}_4\text{mim}]^+[\text{C}_0\text{SO}_4]^-$, $[\text{C}_4\text{mim}]^+[\text{C}_1\text{SO}_4]^-$ and $[\text{C}_4\text{mim}]^+[\text{C}_8\text{SO}_4]^-$ -treated devices: (a) V_{OC} ; (b) FF; (c) J_{SC} ; (d) PCE; (e) SPO and (f) the corresponding J–V curves. The box plot denotes the median (center line), 75th (top edge of the box) and 25th (bottom edge of the box) percentiles. The colored diamond and curves are the statistical data points and corresponding normal distribution curves. All these performance parameters are obtained on the reverse scan from 12 individual devices.

We then proceed to investigate the stabilities of the complete photovoltaic cells incorporated with different surfactant additives under MPPT (Figure R5). For the control device, the PCE quickly decreased to around 70% after roughly 100 hours of aging. By comparison, for the $[\text{C}_4\text{mim}]^+[\text{C}_0\text{SO}_4]^-$ and $[\text{C}_4\text{mim}]^+[\text{C}_1\text{SO}_4]^-$ -treated cells, we observed the roughly 16% and 7% drop in the PCEs for 300 hours aging. Moreover, the early-time ‘burn-in’ was observed in the $[\text{C}_4\text{mim}]^+[\text{C}_0\text{SO}_4]^-$ -treated cell during the MPPT measurement.

Figure R5. The operational stability of the unencapsulated control, $[\text{C}_4\text{mim}]^+[\text{C}_0\text{SO}_4]^-$ and $[\text{C}_4\text{mim}]^+[\text{C}_1\text{SO}_4]^-$ -treated single-junction devices under MPPT in air.

Our Response:

The effects of the surfactant additives consisting of $[\text{C}_n\text{SO}_4]^-$ featuring different alkyl chain lengths on device performance and stability had been added in the Supplementary Figs. 33 and 47. The corresponding discussion had been added in the caption of Supplementary Figs. 33 and 47, respectively.

On page 8, line 313-318, "To evaluate the effects of the surfactant additives consisting of $[\text{C}_n\text{SO}_4]^-$ featuring different alkyl chain lengths as well as

[C₄mim]⁺[BF₄]⁻ at the device level, we also performed J–V and SPO curves of the surfactant additives treated devices as shown in Supplementary Fig. 33." had been added.

On page 10, line 399-402, "Having demonstrated the improved stability of [C₄mim]⁺[C₈SO₄]⁻-containing perovskite devices, we also proceed to investigate the MPPT stabilities of the complete photovoltaic cells incorporated with different surfactant additives as shown in Supplementary Fig. 47." had been added.

6. Comment: The authors claimed ‘reduction of nonradiative recombination’ based on the increased PL lifetime, which is not solid enough. It is suggested to also provide the PLQY measurement results of the films to support the claim.

Our Reply:

Thanks for your precious comments and suggestions. The photoluminescence quantum yield (PLQY) measurements (Figure R6) had been carried out to support the claim of reduction of nonradiative recombination [*Nature* 555, 497-501 (2018).10.1038/nature25989; *Adv Mater* 22, 2208320. 10.1002/adma.202208320]. The PLQY was measured under 532 nm laser excitation at an intensity of 0.5-sun equivalent and increased by ~4 times to 0.39% for the target films compared with 1.61% for the control films (Figure R6). The results quantify the defect-mediated nonradiative recombination of photogenerated charge carriers in perovskite films and hence predict the ultimate device performance.

Figure R6. PLQY values from six individual control films and target films on glass substrates. The PLQY was measured under 532 nm laser excitation at an intensity of 0.5-sun equivalent (24 mW/cm^2) and increased from 1.61% for the control films to 0.39% for the target films. The results quantify the defect-mediated nonradiative recombination of photogenerated charge carriers in perovskite films and hence predict the ultimate device performance. [*ACS Energy Lett* 7, 1903-1911 (2022).10.1021/acseenergylett.2c00865].

Our Response:

The PLQY values from six individual control films and target films on glass substrates had been added in Supplementary Fig. 20.

On page 7, line 242-243, "This can also be confirmed by the results of the photoluminescence quantum yield (PLQY) measurements (Supplementary Fig. 20)." had been added.

7. Comment: The authors provide a detailed explanation of the upshifted VBM, while the upshifted Fermi level was somehow ignored. Considering the important role of the surface properties of perovskite film in p-i-n solar cells, the LAS-induced n-type doping should be further explained.

Our Reply:

Thanks for your constructive comments. It is well known that the E_F shift in the perovskite film originated from a change in the ratio of lead halide- to organic halide-terminated surfaces [*Science* 360, 1442-1446 (2018).

[10.1126/science.aap9282](https://doi.org/10.1126/science.aap9282)], and the energy level of the organic iodide termination is shallower than that of the lead iodide termination [*Chem Mater* 29, 958-968 (2017).[10.1021/acs.chemmater.6b03259](https://doi.org/10.1021/acs.chemmater.6b03259)]. Here, for the target perovskite with the [C₄mim]⁺[C₈SO₄]⁻ additive, the anions [C₈SO₄]⁻ mainly distributed at the top of perovskite film (Supplementary Fig. 9) can bind to lead iodide termination (Supplementary Fig. 16). Therefore, the surface is dominated by the organic halide, indicating a more n-type nature for the target film.

Our Response:

On page 8, line 280-286, "It is well known that the E_F shift in the perovskite film originated from a change in the ratio of lead halide- to organic halide-terminated surfaces, and the energy level of the organic iodide termination is shallower than that of the lead iodide termination. Here, for the target perovskite with the [C₄mim]⁺[C₈SO₄]⁻ additive, the anions [C₈SO₄]⁻ mainly distributed at the top of perovskite film (Supplementary Fig. 9) can bind to lead iodide termination (Supplementary Fig. 16). Therefore, the surface is dominated by the organic halide, indicating a more n-type nature for the target film." had been added.

8. Comment: The authors provided detailed evolution of the performance parameters during the ISOS-L2 measurement, I am wondering if the device hysteresis has also changed with the device degradation. That would be more clear if the authors could provide some scanned J–V curves and the SPO results at different intervals during the measurement.

Our Reply:

Thank you very much for your instructive suggestion. The J–V curves and the SPO results had been summarized in **Figure R7**. The hysteresis index (H-index) was defined based on the following formula [*Nat Energy* 6, 624-632 (2021).[10.1038/s41560-021-00830-9](https://doi.org/10.1038/s41560-021-00830-9)]:

$$\text{Hysteresis index} = \frac{\text{PCE}_{\text{reverse}} - \text{PCE}_{\text{forward}}}{\text{PCE}_{\text{reverse}}}$$

The J–V and the measured SPO curves at different aging times had been presented in **Figure R7**, clearly showing the evolution of device performance—including the increased hysteresis in the J–V curves—during ISOS-L2 aging. We observed a noticeably slow increase of hysteresis in the target device according to the recorded J–V curves in **Figure R7a-c**. Moreover, the SPO values also degrade more quickly than the J–V-determined efficiency (**Figure R7d**), and we observed roughly 12% and 9% SPO decrease in the initial performance after 1,100 hours of aging in the target cells and the champion cell, respectively. By comparison, a roughly 50% drop in the SPO over the 300 hours of aging was observed for the control cells.

Figure R7. J–V curves of (a) control and (b) target devices at different intervals during the ISOS-L2 measurements, in forward scan (dotted lines) and reverse scan (solid lines). (c) The hysteresis index for the corresponding devices during the ISOS-L2 measurements. (d) SPO of the non-encapsulated control and target devices, during aging during ISOS-L2 measurements. The target champion cell

(denoted with*) is indicated with yellow stars (surrounded by black border lines). The standard deviation (error bar) is calculated from six individual devices in the same batch.

Our Response: The J–V curves and the SPO results had been summarized and added in Figure R7,

On page 10, line 384-392, "We also presented the J–V and the measured SPO curves at different aging times in Supplementary Fig. 45, clearly showing the evolution of device performance—including the increased hysteresis in the J–V curves—during ISOS-L2 aging. Besides, we observed a noticeably slow increase of hysteresis in the target device according to the recorded J–V curves. Moreover, the SPO values also degrade more quickly than the J–V-determined efficiency, and we observed roughly 12% and 9% SPO decrease in the initial performance after 1,100 hours of aging (Supplementary Fig. 45) in the target cells and the champion cell, respectively. By comparison, the roughly 50% drop in the SPO over the 300 hours of aging was observed for the control cells." had been added.

9. Comment: There are some additional typos that need to be carefully checked and revised, e.g. “residue stresses”, “Young’s modulus”

Our Reply:

Thanks a lot for pointing out these mistakes.

Our Response:

On page 6, line 220-223, "...and to significantly buffer the residue stresses induced by the volume variation." had been revised to "...and to significantly buffer the residual stresses induced by the volume variation."

On page 2, line 69-72, "...that effectively eliminates stresses by reducing the Young’s modulus and thermal expansion coefficient..." had been revised to "...that effectively eliminates stresses by reducing the Young’s Modulus and thermal expansion coefficient..."

On page 3, line 78-83, "...to decrease the Young’s modulus (YM) and the thermal expansion coefficient (CTE) of the perovskite films..." had been revised to "...to

decrease the **Young's Modulus** (YM) and the thermal expansion coefficient (CTE) of the perovskite films...".

Reviewer #3 (Remarks to the Author):

The authors have introduced a series of long-alkyl-chain anionic surfactant additives to the perovskite precursor, and studies the effect of the alkyl chain length on the perovskite crystallization kinetics and thus the perovskite film optoelectronic and mechanic properties, as well as the as-prepared perovskite and tandem solar cell performance, including both efficiency and stability. The long alkyl chain anionic surfactant is based on $[\text{C}_4\text{mim}]^+[\text{C}_n\text{SO}_4]^-$, for which the cation has been previously introduced to perovskite solar cell (PSC) **but the introduction of the anion part is for the first time being reported, highlighting the novelty of the work.** Longer alkyl chain ($n=8$) was found to be beneficial to reduce the perovskite film residual stress and thus improve the as-prepared device efficiency and stability.

Our Response:

We deeply appreciate the reviewer's approval of the significance of our work. Your professional and constructive comments and suggestions guide us to think about some points more deeply and are very helpful to improve the quality of this manuscript.

- 1. Comment:** This work has performed comprehensive characterisation at both material and device levels and provided sounded and robust data interpretation. However, an important question to be pointed out is the inconsistency between the perovskite film characterisation and the device testing. For perovskite film characterisation, various surfactants with different alkyl chain length ($n=0, 1, 8$) as well as different anion (BF_4^-) were compared to assess their impact of perovskite film quality. However, only one surfactant $[\text{C}_4\text{mim}]^+[\text{C}_8\text{SO}_4]^-$ was tested and reported at the device level. This makes it incomplete to understand the correlation between the film optoelectronic property and the device performance. Hence, it is strongly suggested that all the surfactant molecules are tested at device level. This is particularly important given the fact that the device performance (21%) reported in this work still lags behind that of the state-of-the-art (>24%) with the same bandgap. This raises the question

whether such surfactant could also benefit the state-of-the-art perovskite film and device.

Our Reply:

Thank you for your kind suggestion. The effects of the surfactant additives consisting of $[\text{C}_n\text{SO}_4]^-$ featuring different alkyl chain lengths as well as $[\text{C}_4\text{mim}]^+[\text{BF}_4]^-$ on device performance had been added in the **Figure R8**. The PCE improvements were mainly the result of enhanced V_{OC} and FF in various surfactant-treated devices. The improvement of $[\text{C}_4\text{mim}]^+[\text{C}_8\text{SO}_4]^-$ -treated devices was the most significant, while the $[\text{C}_4\text{mim}]^+[\text{C}_0\text{SO}_4]^-$ and $[\text{C}_4\text{mim}]^+[\text{BF}_4]^-$ -treated devices were accompanied with an obvious hysteresis. The differences in device performance were mainly attributed to the different distribution of additives in perovskite films.

Figure R8. Performance parameters distribution for the control, $[\text{C}_4\text{mim}]^+[\text{C}_0\text{SO}_4]^-$, $[\text{C}_4\text{mim}]^+[\text{C}_1\text{SO}_4]^-$, $[\text{C}_4\text{mim}]^+[\text{C}_8\text{SO}_4]^-$ and $[\text{C}_4\text{mim}]^+[\text{BF}_4]^-$ -treated devices: (a) V_{OC} ; (b) FF; (c) J_{SC} ; (d) PCE; (e) SPO and (f) the corresponding J–V curves. The box plot denotes the median (center line), 75th (top edge of the box) and 25th (bottom edge of the box) percentiles. The colored diamond and curves are the statistical data points and corresponding normal distribution curves. All these performance parameters are obtained on the reverse scan from 12 individual devices.

We also proceed to investigate the stabilities of the complete photovoltaic cells incorporated with different surfactant additives under MPPT (Figure R9). For the control device, the PCE quickly decreased to around 70% after roughly 100 hours of aging. By comparison, for the $[\text{C}_4\text{mim}]^+[\text{C}_0\text{SO}_4]^-$, $[\text{C}_4\text{mim}]^+[\text{C}_1\text{SO}_4]^-$ and $[\text{C}_4\text{mim}]^+[\text{BF}_4]^-$ -treated cells, we observed the roughly 16%, 7% and 10% drop in the PCEs for 300 hours ageing. Moreover, the early-time ‘burn-in’ was observed in the $[\text{C}_4\text{mim}]^+[\text{C}_0\text{SO}_4]^-$ -treated cell during the MPPT measurement.

Figure R9. The operational stability of the unencapsulated control, $[\text{C}_4\text{mim}]^+[\text{C}_0\text{SO}_4]^-$, $[\text{C}_4\text{mim}]^+[\text{C}_1\text{SO}_4]^-$ and $[\text{C}_4\text{mim}]^+[\text{BF}_4]^-$ -treated single-junction devices under MPPT in air.

Indeed, the device performance here is still below the state-of-the-art NiO_x -based (PCE = 23.9%, SPO = 23.9%, $V_{\text{OC}} = 1.15$ V, $J_{\text{SC}} = 24.90$ mA/cm^{-2} , FF = 83.46%, *Nat Photonics* 16, 352-358 (2022).[10.1038/s41566-022-00985-1](https://doi.org/10.1038/s41566-022-00985-1)) and the state-of-the-art p-i-n based (PCE = 25.56%, SPO = 25.5%, $V_{\text{OC}} = 1.208$ V, $J_{\text{SC}} = 25.08$ mA/cm^{-2} , FF = 84.37%, and a certified steady-state efficiency of 24.7%, *Science* 379, 683-690 (2023). [10.1126/science.ade3126](https://doi.org/10.1126/science.ade3126)) perovskite solar cells. Despite that, Performance of our device (PCE = 21.6%; SPO = 21.4%) is still comparable to the state-of-the-art device with a bandgap of 1.63 eV, for example, PCE of 21.8% (SPO of 21.6%) for [*Science* 354, 206-209 (2016). [10.1126/science.aah5557](https://doi.org/10.1126/science.aah5557)], PCE of 19.8% for [*Nat Energy* 7, 744-753 (2022).[10.1038/s41560-022-01076-9](https://doi.org/10.1038/s41560-022-01076-9)], PCE of 21.5% (SPO of 21.3%) for [*Nat Photonics* 15, 681-689 (2021).[10.1038/s41566-021-00829-4](https://doi.org/10.1038/s41566-021-00829-4)], PCE of 22.1% (SPO of 21.7%) for [*Adv Mater* 34, 2106280 (2022).[10.1002/adma.202106280](https://doi.org/10.1002/adma.202106280)], PCE of 21.0% (SPO of 21.0%) for [*Energy Environ Sci* 14, 3976-3985 (2021).[10.1039/D0EE03807E](https://doi.org/10.1039/D0EE03807E)],

PCE of 20.3% (SPO of 20.5%) for [*Adv Energy Mater* 13, 2203313 (2023).10.1002/aenm.202203313], PCE of 20.3% for [*Nano Energy* 109, 108268 (2023).10.1016/j.nanoen.2023.108268], PCE of 20.4% for [*Nano Energy* 86, (2021).10.1016/j.nanoen.2021.106114], PCE of 21.8% (SPO of 21.1%) for [*Adv Funct Mater* 32, 2200431 (2022).10.1002/adfm.202200431], PCE of 22.3% (SPO of 22.0%) for [*ACS Appl Mater Interfaces* 13, 13022-13033 (2021).10.1021/acsaami.0c17893], PCE of 21.5% (SPO of 20.5%) for [*Mater Adv* 3, 5786-5795 (2022).10.1039/D2MA00391K]. The key photovoltaic parameters for high-performance perovskite solar cells with a bandgap of 1.63 eV had also been summarized in **Table R1**. The efficiency loss in our work is mainly due to the lack of passivation at the NiO_x/perovskite and perovskite/C₆₀ interfaces. Further improvements of the device performance will be studied in our future work.

Table R1. Summary of key photovoltaic parameters for high-performance perovskite solar cells with the bandgap of 1.63 eV.

Sample	Date	V _{oc} (V)	FF (%)	J _{sc} (mA/cm ²)	PCE (%)	SPO (%)
Saliba M, et al. (Science)	2016.08.29	1.18	81.0	22.8	21.8	21.6
Dagar J, et al. ACS Appl. Mater. Interfaces	2021.03.15	1.18	83.3	22.7	22.3	22.0
Liu X, et al. Nano Energy	2021.05.04	1.13	81.5	22.1	20.4	--
Aktas E, et al. Energy Environ. Sci.	2021.05.21	1.16	80.0	22.6	21.0	21.0
Yang G, et al. Nat. Photonics	2021.07.05	1.24	77.6	22.4	21.5	21.3
Li Y, et al. Adv. Mater.	2021.12.09	1.24	77.8	22.9	22.1	21.7
Zheng Y, et al. Adv. Funct. Mater.	2022.04.24	1.18	81.8	22.6	21.8	21.1
Jiang J, et al. Mater. Adv.	2022.06.13	1.20	82.4	21.7	21.5	20.5
Wang C, et al. Nat. Energy	2022.07.21	1.15	77.8	22.2	19.8	--
Li B, et al. Adv. Energy Mater.	2022.10.30	1.20	83.0	22.1	21.9	21.3

Almora O, et al. Adv. Energy Mater.	2022.11.07	1.13	76.8	23.4	20.3	20.5
Castriotta LA, et al. Nano Energy	2023.02.08	1.14	83.2	21.5	20.3	--
This work		1.13	82.7	23.1	21.6	21.4

Our Response:

On page 8, line 313-318, "To evaluate the effects of the surfactant additives consisting of $[C_nSO_4]^-$ featuring different alkyl chain lengths as well as $[C_4mim]^+[BF_4]^-$ at the device level, we also performed J–V and SPO curves of the surfactant additives treated devices as shown in Supplementary Fig. 33." had been added.

On page 10, line 399-402, "Having demonstrated the improved stability of $[C_4mim]^+[C_8SO_4]^-$ -containing perovskite devices, We also proceed to investigate the MPPT stabilities of the complete photovoltaic cells incorporated with different surfactant additives as shown in Supplementary Fig. 47." had been added.

The effects of the surfactant additives consisting of $[C_nSO_4]^-$ featuring different alkyl chain lengths as well as $[C_4mim]^+[BF_4]^-$ on device performance and stability had been added in the Supplementary Figs. 33 and 47.

The discussion: "Compared with control devices, the device statistics corroborated that the PCE improvements were mainly the result of enhanced V_{OC} and FF in the treated devices. The improvement of $[C_4mim]^+[C_8SO_4]^-$ -treated devices is the most significant, while the $[C_4mim]^+[C_0SO_4]^-$ and $[C_4mim]^+[BF_4]^-$ -treated devices are accompanied with an obvious hysteresis. The differences in device performance are likely attributed to the different distribution of additives in perovskite films." had been added in the caption of Supplementary Figs. 33.

The discussion: " For the control device, the PCE quickly decreased to around 70% after roughly 100 hours of aging. By comparison, for the $[C_4mim]^+[C_0SO_4]^-$, $[C_4mim]^+[C_1SO_4]^-$ and $[C_4mim]^+[BF_4]^-$ -treated cells, we observed the roughly 16%, 7% and 10% drop in the PCEs for 300 hours aging. Moreover, the early-time

'burn-in' was observed in the $[\text{C}_4\text{mim}]^+[\text{C}_6\text{SO}_4]^-$ -treated cell during the MPPT measurement." had been added in the caption of Supplementary Figs. 47.

On page 8, line 313-315, "The performance of our device is comparable to the state-of-the-art device with the bandgap of 1.63 eV (Supplementary Table 4)." had been added.

2. Comment: Another technical question to answer is that the authors also observed increased YMs for short-chain $[\text{C}_4\text{mim}]^+[\text{BF}_4]^-$ additive. This seems to be contradictory to the report by Bai, S. et al [Ref. 30, Planar perovskite solar cells with long-term stability using ionic liquid additives. *Nature* 571, 245-250 (2019).], where $[\text{C}_4\text{mim}]^+[\text{BF}_4]^-$ was found to improve the PSC efficiency and stability. Therefore, in this work, the authors are suggested to also test the device performance for the PSC treated with $[\text{C}_4\text{mim}]^+[\text{BF}_4]^-$ for comparison.

Our Reply:

Thanks for the helpful comments which guide us to make this point clearer. The J-V and SPO results of the device based on $[\text{C}_4\text{mim}]^+[\text{BF}_4]^-$ are shown in Figure R10 and Figure R11, respectively. The $[\text{C}_4\text{mim}]^+[\text{BF}_4]^-$ -treated devices exhibited an increased PCE due to the enhanced V_{OC} and FF compared with the control devices (Figure R10) but showed a decreased PCE compared with the target samples. Meanwhile, we also observed an increased hysteresis for the $[\text{C}_4\text{mim}]^+[\text{BF}_4]^-$ -treated device, which can be explained by the accumulation of $[\text{BF}_4]^-$ at the bottom of the perovskite film [*Nature* 571, 245-250 (2019).10.1038/s41586-019-1357-2]. It is well known that both defect passivation and stress reduction can synergistically contribute to the device performance. To further quantify the long-term stability, we also performed MPP tracking experiment under simulated full-sun irradiance (Figure R12). The $[\text{C}_4\text{mim}]^+[\text{BF}_4]^-$ -treated device showed a better stability compared to the control device, but was less stable than the $[\text{C}_4\text{mim}]^+[\text{C}_8\text{SO}_4]^-$ -treated device. The improved stability compared to the control device is due to more effective defect passivation in the

$[\text{C}_4\text{mim}]^+[\text{BF}_4]^-$ -treated device, while the poor stability compared to the $[\text{C}_4\text{mim}]^+[\text{C}_8\text{SO}_4]^-$ -treated device is caused by the increased stress in the $[\text{C}_4\text{mim}]^+[\text{BF}_4]^-$ -treated device.

Figure R10. J–V curves of the best-performing $[\text{C}_4\text{mim}]^+[\text{BF}_4]^-$ -treated device, in forward scan (dotted lines) and reverse scan (solid lines). The device showed a notable PCE of 20.9% (20.4%) with an FF of 81.5% (79.8%), a V_{OC} of 1.12 V (1.12 V) and a J_{SC} of $22.9 \text{ mA}\cdot\text{cm}^{-2}$ ($22.8 \text{ mA}\cdot\text{cm}^{-2}$) under the reverse (forward) voltage scan.

Figure R11. SPO of the $[\text{C}_4\text{mim}]^+[\text{BF}_4]^-$ -treated device. The V_{max} and J_{max} are 0.96 V and $21.6 \text{ mA}\cdot\text{cm}^{-2}$, respectively.

Figure R12. The operational stability of the unencapsulated $[C_4mim]^+[BF_4]^-$ -treated single-junction device under MPPT in air.

Our Response:

The results of J–V, SPO and MPPT measurements had been added in Supplementary Figs. 33 and 47. The corresponding discussion had been added in the caption of Supplementary Figs. 33 and 47, respectively.

3. Comment: The fabrication process of how to introduce the surfactant into the perovskite film is not clear. In the experimental part, for the single-junction PSC, the description of “...the surfactant-containing precursor solutions were prepared by dissolving the same $CS_{0.05}(FA_{0.83}MA_{0.17})_{0.95}Pb(I_{0.82}Br_{0.18})_3$ perovskite components in the desired molar ratios surfactant-containing DMF/DMSO mixed solvent, and all solutions were filtered (0.45 μ m, 431 PTFE) before use” does not read clear. For the tandem solar cell, no description about how to incorporate the surfactant can be found.

Our Reply:

Sorry for the unclear expression and thanks for your valuable suggestions. The surfactant-containing precursor solutions were prepared according to the following methods. First, the surfactant was dissolved in a mixed solvent (4:1 in volume) of DMF/DMSO with the desired molar concentration. Then, the surfactant-containing precursor solutions were prepared by dissolving the perovskite components in the surfactant-containing DMF/DMSO mixed solvent, and all solutions were filtered (0.45 μ m, PTFE) before use. For surfactant-

containing tandem solar cells, the preparation of the surfactant-containing precursor solutions is the same as for single-junction solar cells.

Our Response:

On page 12, line 464-468, "...the surfactant-containing precursor solutions were prepared by dissolving the same $\text{Cs}_{0.05}(\text{FA}_{0.83}\text{MA}_{0.17})_{0.95}\text{Pb}(\text{I}_{0.82}\text{Br}_{0.18})_3$ perovskite components in the desired molar ratios surfactant-containing DMF/DMSO mixed solvent, and all solutions were filtered (0.45 μm , 431 PTFE) before use" had been revised to "In parallel, the surfactant was dissolved in a mixed solvent (4:1 in volume) of DMF/DMSO with the desired molar concentration. Then, the surfactant-containing precursor solutions were prepared by dissolving the perovskite components in the surfactant-containing DMF/DMSO mixed solvent, and all solutions were filtered (0.45 μm , PTFE) before use."

On page 13, line 502-504, "For surfactant-containing precursor solutions, the preparation of the solutions is the same as for single-junction solar cells." had been added to describe how to incorporate the surfactant for the tandem solar cells.

4. Comment: The light stability of the tandem solar cell was reported to be much worse than that of the single-junction PSC. The authors pointed out one potential reason which is "the free carriers accumulated at the poor ETL/perovskite interface will reduce the ion migration activation energy and then accelerate the perovskite degradation," Is there any evidence for this hypothesis? In addition, there is prominent current mismatch between the Si sub-cell and the perovskite sub-cell according to the EQE of the tandem device. Can the authors comment on the impact of the current mismatch on the stability of the tandem device?

Our Reply:

We clearly recognize the concern raised by the reviewer. On one hand, more photoexcited charge carriers are present at the ETL interface for the tandem device. As we know, the illumination is incident from the HTL(NiO_x)/perovskite side for the single-junction device, and from the perovskite/ETL(C_{60}) side for the tandem

device. Therefore, most charge carriers generate near the HTL/perovskite interface for the single-junction device but inversely near the perovskite/ETL interface for the tandem device. On the other hand, the defect density at the front ETL side are larger than the bottom HTL side [*Energy Environ Sci* 14, 1563-1572 (2021).10.1039/D1EE00116G]. It is well known that the C₆₀ interface is much poorer than the HTL interface, making the perovskite/ETL interface more prone to interface defect density than the perovskite/HTL layer interface. Therefore, the high-density of photoexcited charge carriers together with the high-density of defects at the perovskite/ETL side can accelerate ion migration and hinder the stability of the tandem devices. [*Adv Mater* 31, 1902413 (2019). 10.1002/adma.201902413; *Nat Mater* 17, 445-449 (2018).10.1038/s41563-018-0038-0; *J Am Chem Soc* 140, 1358-1364 (2018).10.1021/jacs.7b10430; *Nat Commun* 9, 4981 (2018).10.1038/s41467-018-07438-w]. To demonstrate the accumulation of carriers at the ETL side, drive-level capacitance profiling (DLCP) [*Joule* 4, 1949-1960 (2020). 10.1016/j.joule.2020.07.003] had been performed and the results are shown in **Figure R13**. Here, by measuring the device at 1 MHz in the dark, we observed that the carrier concentration in the region closer to the C₆₀/perovskite interface increased by up to 2 times compared with perovskite/NiO_x, indicating a more serious free carriers accumulation at the poor ETL/perovskite interface. Similarly, since the perovskite top cell here is not the limiting cell and thus the photoexcited charge carriers in the perovskite layer are not all extracted, the long-term stability of the PVK-based tandems could be affected this current mismatch [*Sol RRL* 5, 2100311 (2021). 10.1002/solr.202100311]. The impact of the current mismatch on the stability of the tandem device will be studied in our future work.

Figure R13. The distribution of carriers in the perovskite layer obtained by DLCP at 1 MHz.

Our Response:

On page 11, line 430-433, we had added "Similarly, since the perovskite top cell here is not the limiting cell and thus the photoexcited charge carriers in the perovskite layer are not all extracted, the long-term stability of the perovskite-based tandems could be affected this current mismatch, and may be resolved in the future study."

On page 15, line 588-589, "The drive-level capacitance profiling (DLCP) was measured by an Agilent E4980A." had been added.

The result of DLCP measurement and the corresponding information had been added in Supplementary Figs. 56.

REVIEWERS' COMMENTS

Reviewer #1 (Remarks to the Author):

The authors have addressed the issues and concerns in the review satisfactorily. The novel and thorough work (vast suite of characterization techniques including in-depth explanations) on the LAS additives for perovskite/silicon tandem solar cells would be well received by the perovskite scientific community. Thus, the manuscript should be suitable for publication, provided the concerns from the other reviewers have been adequately addressed as well.

Minor correction:

Line 255: “overserved” should be “observed” here.

Reviewer #2 (Remarks to the Author):

In the revised version, the authors provided additional solid results on the thin film, device performance, and device stability results for samples containing different LASs. These new results have properly addressed most of my concerns and further improved the quality of the manuscript. I believe that the presented new design of the anion part of the LSAs, the comprehensive characterization and understanding of the associated mechanisms on the stability improvement would be also insightful for further developments of both single-junction and tandem perovskite solar cells. I have no further questions and recommend acceptance of the manuscript in its current version.

Reviewer #3 (Remarks to the Author):

In the revised version of the manuscript, the authors have adequately addressed all the questions raised during the first round of review. They have checked the device performance by testing solar cell performance using different additives, conducted device stability testing, and aimed to understand the difference in stability between single-junction and tandem devices.

Based on the authors' responses and the revised manuscript, I suggest publishing this work in Nature Communications.

List of point-to-point response of reviews' comments

Reviewer #1 (Remarks to the Author):

The authors have addressed the issues and concerns in the review satisfactorily. The novel and thorough work (vast suite of characterization techniques including in-depth explanations) on the LAS additives for perovskite/silicon tandem solar cells would be well received by the perovskite scientific community. Thus, the manuscript should be suitable for publication, provided the concerns from the other reviewers have been adequately addressed as well.

Our Response:

We deeply appreciate the reviewing efforts and positive comments.

Minor correction:

Line 255: “overserved” should be “observed” here.

Our Reply:

Thanks a lot for pointing out this mistake.

Our Response:

On page 7, line 253-256, "...between the GBs and GIs can be overserved for the target perovskite surface..." had been revised to "...between the GBs and GIs can be observed for the target perovskite surface...".

Reviewer #2 (Remarks to the Author):

In the revised version, the authors provided additional solid results on the thin film, device performance, and device stability results for samples containing different LASs. These new results have properly addressed most of my concerns and further improved the quality of the manuscript. I believe that the presented new design of the anion part of the LSAs, the comprehensive characterization and understanding of the associated mechanisms on the stability improvement would be also insightful for further developments of both single-junction and tandem perovskite solar cells. I have no further questions and recommend acceptance of the manuscript in its current version.

Our Response:

We thank the referee for a constructive review process.

Reviewer #3 (Remarks to the Author):

In the revised version of the manuscript, the authors have adequately addressed all the questions raised during the first round of review. They have checked the device performance by testing solar cell performance using different additives, conducted device stability testing, and aimed to understand the difference in stability between single-junction and tandem devices.

Based on the authors' responses and the revised manuscript, I suggest publishing this work in Nature Communications.

Our Response:

We express our gratitude to the referee for the constructive review process.